# Endothelialization of arterial vascular grafts by circulating monocytes

Randall J. Smith Jr.[1], Bita Nasiri[2], Julien Kann[3], Donald Yergeau [3], Jonathan E. Bard [3], Daniel D. Swartz[4] & Stelios T. Andreadis[1,2,4,5 ✉]

Recently our group demonstrated that acellular tissue engineered vessels (A-TEVs) comprised of small intestinal submucosa (SIS) immobilized with heparin and vascular endothelial growth factor (VEGF) could be implanted into the arterial system of a pre-clinical ovine animal model, where they endothelialized within one month and remained patent. Here we report that immobilized VEGF captures blood circulating monocytes (MC) with high specificity under a range of shear stresses. Adherent MC differentiate into a mixed endothelial (EC) and macrophage (Mφ) phenotype and further develop into mature EC that align in the direction of flow and produce nitric oxide under high shear stress. In-vivo, newly recruited cells on the vascular lumen express MC markers and at later times they co-express MC and EC-specific proteins and maintain graft patency. This novel finding indicates that the highly prevalent circulating MC contribute directly to the endothelialization of acellular vascular grafts under the right chemical and biomechanical cues.

[1] Department of Biomedical Engineering, University at Buffalo, State University of New York, Amherst, NY 14260, USA. [2] Department of Chemical and Biological, University at Buffalo, State University of New York, Amherst, NY 14260, USA. [3] Genomics and Bioinformatics Core, University at Buffalo, State University of New York, Amherst, NY 14260, USA. [4] Angiograft LLC, Amherst, NY 14260, USA. [5] Center of Excellence in Bioinformatics and Life Sciences, Buffalo, NY 14263, USA. ✉email: sandread@buffalo.edu

Acellular vascular grafts continue to show great promise in various animal models as well as human clinical trials. Decellularized tissue engineered constructs have been utilized with increasing frequency and demonstrated improved patency and regeneration potential in pre-clinical studies[1–13] and clinical trials[14,15]. In addition to decellularized grafts, non-biological grafts composed of various polymeric biomaterials have also been used to engineer cell-free vascular grafts[16–24]. All of these acellular materials must promote endothelialization of the lumen to achieve patency and promote development of the vascular wall through extensive, long-term remodeling. However, despite extensive experience with engineering vascular grafts, the mechanism of endothelialization, remains unknown.

Rapid endothelialization has been reported in small animal models as well as in pigs and dogs, which occurred mostly via migration of endothelial cells from the anastomotic sites. However, trans-anastomotic endothelialization is very limited in ovine or humans[25,26] and therefore, the mechanism of endothelialization of acellular vascular grafts remains unclear. Several studies employed immobilized peptides or growth factors to the luminal surface to promote endothelialization. Growth factors such as stromal derived factor (SDF1α) was used to home circulating stem cells to the graft lumen. However, these studies showed incomplete endothelialization, especially in the center of the grafts[27,28]. In our lab we developed an acellular vascular graft that was based on small intestinal submucosa (SIS) with immobilized heparin and VEGF on the graft lumen to capture VEGF receptor expressing cells from the blood. When implanted into the abdominal aorta of a mouse model, the 1-mm diameter VEGF grafts were fully endothelialized within 1 month, consisted of pro-regenerative-anti-inflammatory cells and exhibited distinct vascular remodeling toward the native state[29]. Furthermore, when implanted into the carotid arteries of a clinically relevant ovine animal model, such small diameter (4.5 mm), 5-cm-long grafts exhibited high patency rates, fully endothelialized within 1 month, and developed a functional and contractile medial layer by 3 months post-implantation[30–32].

Given the success of VEGF-based vascular grafts and the lack of trans-anastomotic migration in the ovine animal model, we sought to determine the mechanism of rapid endothelialization of otherwise a-cellular grafts occur given the clear role inflammation plays in the mouse model. Herein we identify a novel means of endothelialization via the capture and subsequent differentiation of circulating monocytes (MC) on the VEGF coated lumen. We show that VEGF captures MC, which significantly outnumber endothelial progenitor cells (EPCs) in the blood, and differentiate into functional EC that produces nitric oxide and affords patency to neo-arteries.

## Results

**Cells of mixed EC and M2 macrophage phenotype populate vascular grafts**. At 1-week post-implantation, the lumen of VEGF functionalized SIS grafts contained cells that were devoid of endothelial cell (EC) markers, such as CD144 and eNOS. In addition, by 1 month the lumen was completely populated with cells and there was no gradient of cell density between the anastomotic sites and the middle of the graft. These observations prompted us to hypothesize that the lumen might be endothelialized with cells from circulating blood.

To this end, the phenotype of luminal cells was assessed by immunocytochemistry. Interestingly, at 1-week post-implantation graft lumens were comprised of CD14+ and CD163+ cells, but lacked EC markers CD144 or eNOS (Fig. 1a, b). Surprisingly, at 1 and 3 months post-implantation, luminal cells of explanted grafts co-expressed the endothelial specific marker CD144 and the M2-macrophage specific marker CD163 (Fig. 1a). Similarly, they also co-expressed the EC-specific marker, eNOS and the monocyte/macrophage specific marker, CD14 (Fig. 1b).

**VEGF captures monocytes from whole blood under flow**. These results prompted us to hypothesize that the lumen of VEGF decorated grafts might be endothelialized with monocytes from circulating blood. To address this hypothesis, we employed a microfluidic device to examine whether cells from blood could be captured by immobilized VEGF. The device consisted of a single channel (length: 1 cm; width: 400 μm; height: 200 μm) that was kept in place by vacuum (Fig. 2a) and coated with a layer of chitosan (positively charged) that was used as adhesive to immobilize heparin (negatively charged) as shown previously[33]. VEGF was then immobilized by binding to heparin via its heparin binding domain. The surface concentration of VEGF increased with increasing the VEGF concentration in solution until saturation was reached at ~1000 ng/cm$^2$ (Fig. 2b). Furthermore, the chitosan/heparin/VEGF (denoted as CHV) surfaces supported proliferation of human umbilical vein endothelial cells (HUVEC) in a VEGF surface concentration dependent manner (Fig. 2c), demonstrating that immobilized VEGF on CHV surface is biologically active.

Next, freshly drawn human blood was passed over the VEGF-containing channel surface under different flow rates corresponding to shear stress ranging from 1 to 15 dyn/cm$^2$ (Fig. 2d) and captured cells were fixed and assessed via immunocytochemistry. As indicated in Fig. 2e (representative images at shear stress of 1 dyn/cm$^2$) and quantified in Fig. 2f, all captured cells expressed the monocyte (MC) marker CD14, independent of the level of shear stress (e.g. 100% of 234 ± 16 cells at 15 dyn/cm$^2$; $n = 10$ independent runs per shear stress were tested). In comparison <1% of captured cells expressed the EC marker CD144 (only one out of 234 ± 16 cells were positive; $n = 5$ independent runs at 1 dyn/cm$^2$ or 15 dyn/cm$^2$; no CD144+ cells were present in any run at shear stress of 5 or 10 dyn/cm$^2$). Furthermore, between 1 and 10 dyn/cm$^2$, all captured cells expressed CD31, a shared marker between MC and EC lineages. Interestingly, 64.3 ± 7.1% of captured cells expressed the anti-inflammatory macrophage marker CD163. These results suggested that VEGF may be capturing blood MC, which are known to express the VEGF receptor 1 (VEGFR1).

To verify any potential MC capture by surface immobilized VEGF, CD14+/CD16+ MC were isolated from peripheral human blood by negative selection and run over the microfluidic channel (0.5 × 10$^6$ cells/mL) under the same shear conditions. Indeed, MC were captured by surface bound VEGF with capture efficiency similar to that of human EC, ovine EC, and a murine macrophage cell line (Fig. 2g; $n = 10$ independent runs at 1 dyn/cm$^2$). In contrast, little or no capture was observed when using murine and human fibroblasts or human mesenchymal stem cells.

**Immobilized VEGF captures MC from peripheral blood mononuclear cells**. Next, we examined the phenotype of VEGF captured MC using multi-color flow cytometry. As a control we used FN-coated surface that has been previously used to culture MC. To this end, we performed flow cytometry of peripheral blood mononuclear cells (PBMNs) directly after standard histopaque-1077 isolation (Fig. 3a) or following adherence on FN (Fig. 3b) or VEGF functionalized surface (Fig. 3c) for 1 h before gentle mechanical removal. As indicated in Fig. 3, the majority of cells (>98 ± 2.7%) that bound to FN or VEGF coated surfaces were positive for CD14, a well-accepted MC marker, as compared to only 29 ± 4.6% CD14+ in the starting PBMNC population. In addition, FN or VEGF-captured MC were highly enriched in classical (CD14++/CD16−, FN: 71 ± 3.9%; VEGF: 73 ± 4.2%)

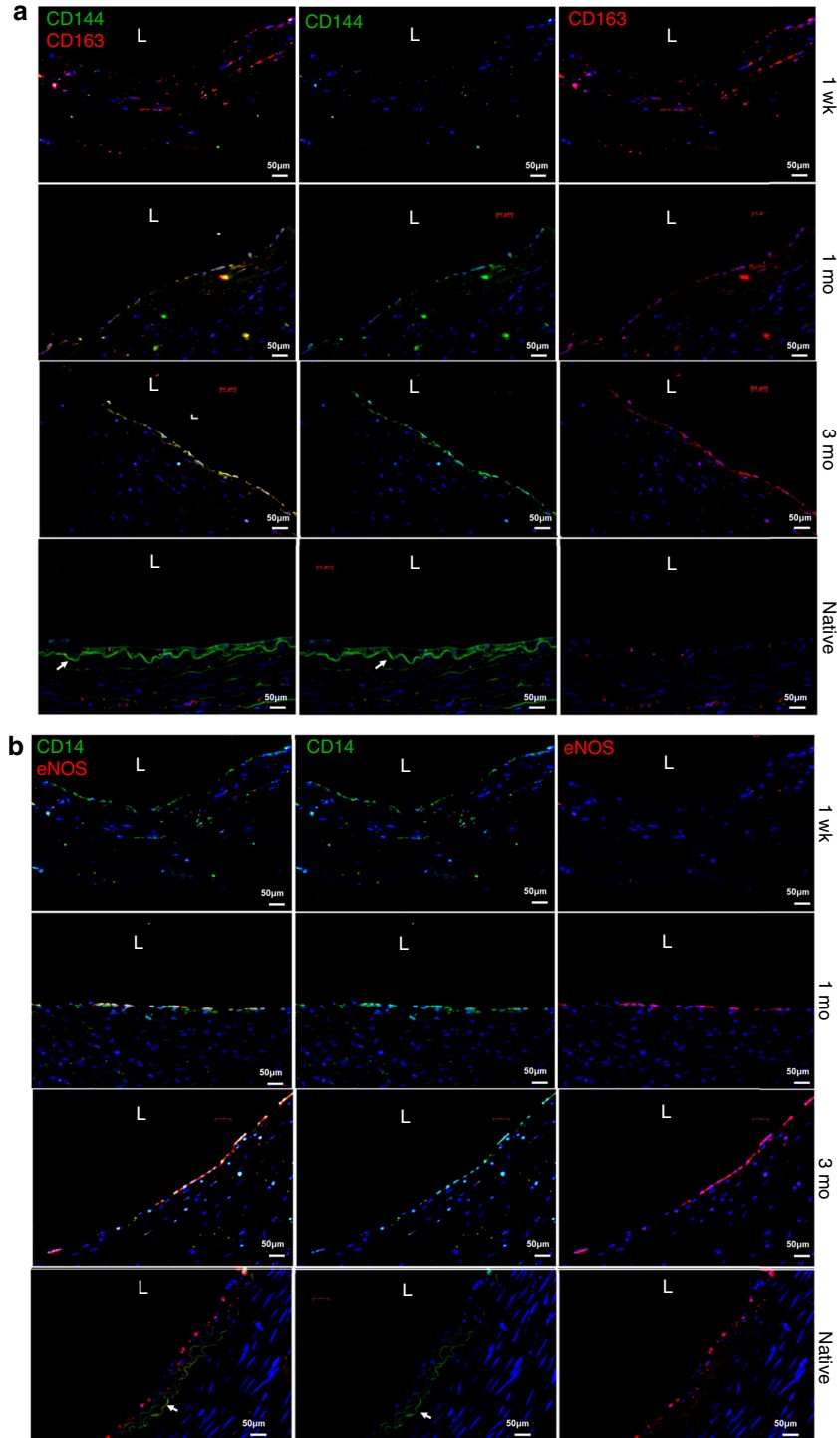

**Fig. 1 Monocytes are incorporated as the endothelium of acellular vascular grafts.** Immunostaining of explanted VEGF-based A-TEVs at the indicated time points of 1 week, 1 month, and 3 months compared to native carotid artery. **a** Co-staining for the macrophage marker CD163 (red) and the endothelial marker CD144 (green). **b** Co-staining for the monocyte marker CD14 (red) and the functional endothelial marker eNOS (green). Note that at 1-week lumens are devoid of EC markers but express MC markers, CD14 and CD163. At 1 and 3 months lumens comprise of cells co-staining for MC (CD14, CD163) and EC markers (CD144, phosphorylated (active) eNOS). White letter "L" indicates the lumen. Scale bars 50 µm. White arrows indicate auto-fluorescent inner elastic lamina.

and non-classical MC (CD14+/CD16+, FN: 28.5 ± 2.1%; VEGF: 26 ± 3.4%). As expected, VEGF-captured MC were highly enriched in VEGFR1 expressing cells (99 ± 1.2% were VEGFR1+), as were the FN bound cells, albeit to a lesser extent (73 ± 5.6%). However, no VEGFR2+ cells were found on either the VEGF or FN coated surface beyond background noise (~2%). Histograms

for each antibody and corresponding IgG controls are shown in Fig. 3d–g. IgG gating is depicted in Supplementary Fig. 1.

**Inducing spreading and proliferation of surface captured MC.** Next, we examined whether MC that were attached on VEGF or FN could be differentiated into functional EC similar to what was

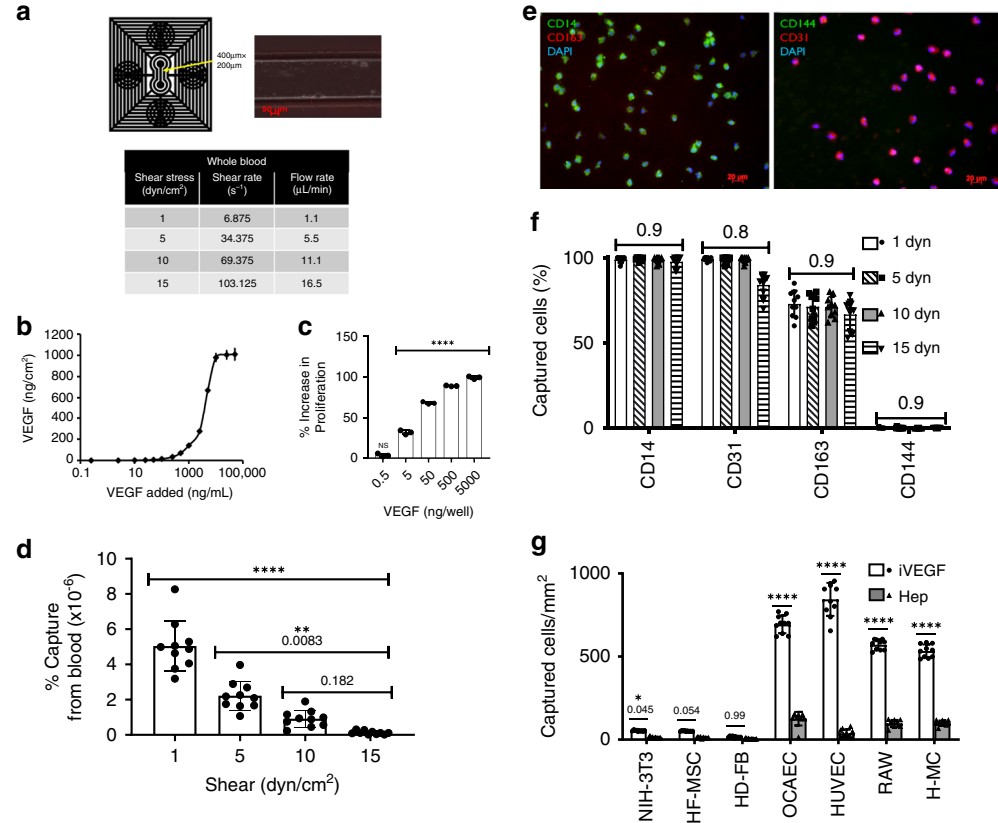

**Fig. 2 VEGF Captures Monocytes Under Flow.** Microfluidic device to capture of cells from blood: schematic and shear dependency. **a** Schematic of microfluidic device. PDMS device is sealed to iVEGF using a vacuum. An empty channel image is shown. **b** VEGF binding kinetics to CH surface as determined by ELISA, $n = 3$ independent biological replicates; error bars denote ± SD of the mean. **c** MTT proliferation of HUVECs on iVEGF with a range of initial soluble VEGF concentrations added per well; $n = 3$ independent biological replicates; Four asterisk indicates statistical difference ($p < 0.0001$) compared to control (no VEGF) using one-way ANOVA and Dunnett's test (DF = 12, F = 2101); error bars denote ±SD of the mean. **d–f** Whole human blood was passed through the microfluidic device on iVEGF at the indicated shear stress. All runs were 1 mL of blood. Total captured cells (**d**) were analyzed as captured cells per total cell count of 1 mL of blood, $n = 10$ independent biological replicates per shear strength; four asterisk indicates statistical difference ($p < 0.0001$), $p$-values as indicated determined by one-way ANOVA and Tukey's test (DF = 36; F = 63.78); error bars denote ±SD of the mean. **e**, **f** Immunostaining with antibodies against MC markers CD14, CD163, and CD31 and EC marker CD144. **e** Representative images at 1 dyn/cm[2] are shown. Images were quantified (**f**) as shown over the 10 independent biological replicates per shear (1 dyn; white bars, 5 dyn; diagonal thatched bars, 10 dyn; gray bars, 15 dyn horizontal thatched bars). No significant difference ($p$ values indicated) between shear strengths and identity using two-way ANOVA and Sidak's test; error bars denote ±SD of the mean. EC markers are observed in <1% of total cells under all shears tested. **g** VEGF captures cells expressing the VEGF receptors with high specificity, $n = 10$ independent biological runs per cell type (NIH-3T3; mouse fibroblast line, HF-MSC human hair follicle-derived mesenchymal stem cells, HD-FB human dermal fibroblasts, HUVEC human umbilical vein endothelial cell, OCAEC ovine carotid artery endothelial cell, RAW mouse macrophage line, H-MC human peripheral blood monocytes). Significance determined using two-way ANOVA and Sidak's test (DF = 36, F = 1.458), significance as indicated; four asterisk denotes statistical difference $p < 0.0001$ as compared to cells captured on CH surface; error bars denote ±SD of the mean; iVEGF white bars, Hep surface gray bars.

observed on the vascular graft lumens in vivo. To this end, we developed a protocol based on two important findings (Fig. 4a). First, we found that the ROCK inhibitor, Y-27632 induced rapid adherence and spreading on both iVEGF and FN (Fig. 4b, c). Quantification of cell area indicates that Y-27632 has a dramatic effect on spreading, with an average cell area of 1126.2 ± 43.8 μm[2] (mean ± SD) and 1006.4 ± 13.7 μm[2] on FN and iVEGF when cultured with Y-27632 (50 nM for 3 days) as compared to 443.9 ± 33.4 μm[2] and 331.0 ± 22.1 μm[2] on FN and iVEGF when cultured without Y-27632 (Fig. 4c).

Also, it is well documented that primary MC are non-dividing cells when cultured in vitro. Interestingly, we discovered that Wnt activation using the GSK3β antagonist, CHIR-99021 (CHIR), induced proliferation of adherent MC, as shown by immunostaining for Ki67 (Fig. 4d). Specifically, treatment with CHIR for 2 days induced Ki67 expression in 44.4 ± 11.6% ($p < 0.05$, $n = 3$)

of cells on FN and 50.2 ± 11.6% ($p < 0.05$, $n = 3$) of cells on VEGF, as compared to only 6.3 ± 4.5% of Ki67+ of control cells on FN (no CHIR) (Fig. 4e).

**Monocyte differentiation to endothelial cells.** Based on these observations, we developed an optimized differentiation protocol as outlined above in Fig. 4a. Briefly, monocytes were cultured on either FN or iVEGF in a modified endothelial basal media (EBM; Lonza) with all supplements (EGM2 bullet kit) with Y-27632 (10 μM), MCSF (10 ng/mL), soluble VEGF (100 ng/mL), and 10% autologous activated platelet rich plasma (PRP). On day 3, Y-27632 was removed and CHIR was added for 2 days. Thereafter, the medium was replaced with the same basal medium but without Y-27632 or CHIR until day 14. While initially cultured monocytes displayed a spindle like morphology, they gradually

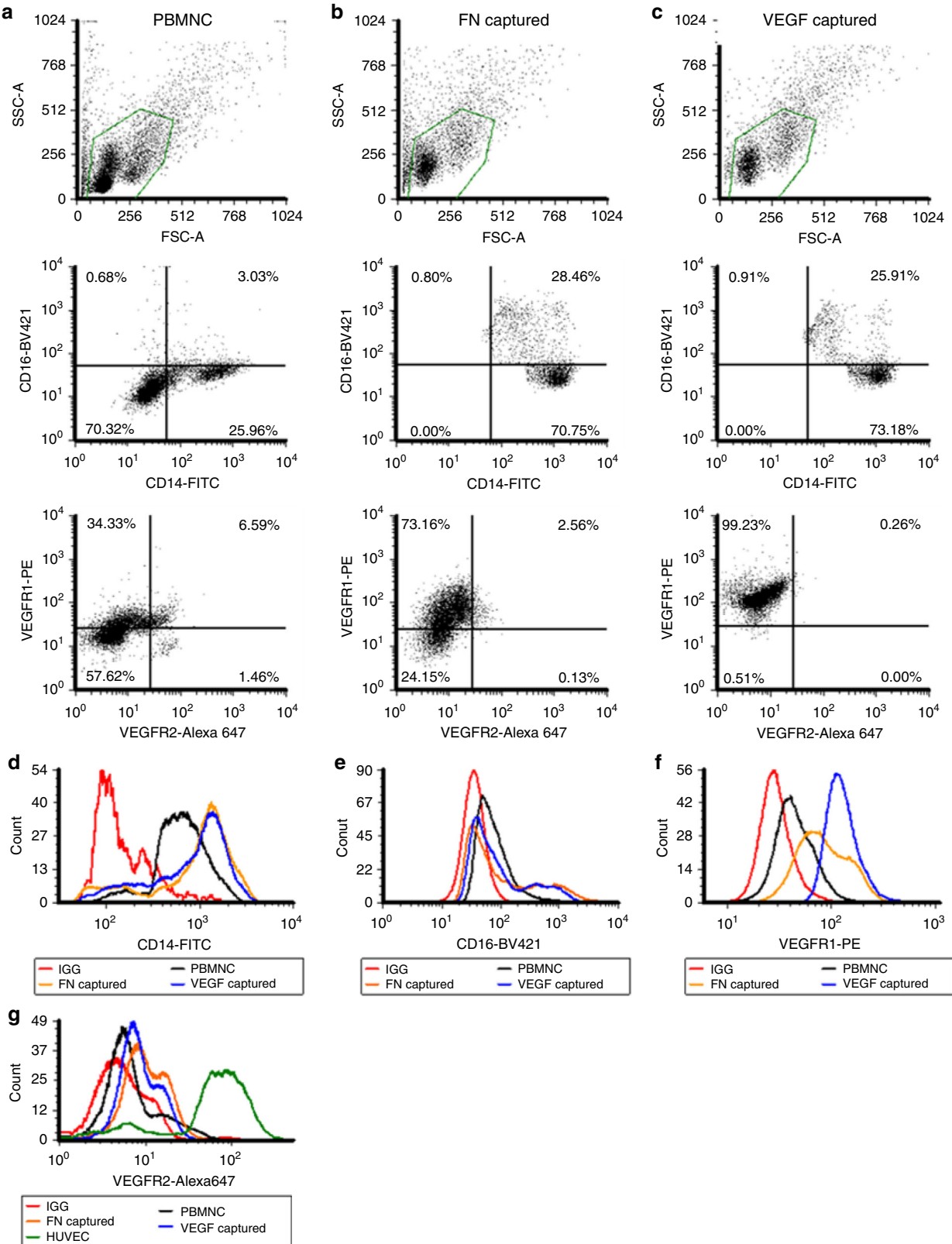

**Fig. 3 Characterization of PBMNs from whole blood.** Multi-color flow cytometry assessment of PBMNCs **a** isolated directly from whole blood; or **b** after 1 h capture on FN surface; or **c** after 1 h capture on iVEGF. Cells were only gated by FSC/SSC (green gate), no sub-gates were used in order to show the entire population of cells adhered to the surfaces. Flow cytometry histograms demonstrating positive signal shifts (**d**–**g**). HUVEC served as a positive control for VEGFR2.

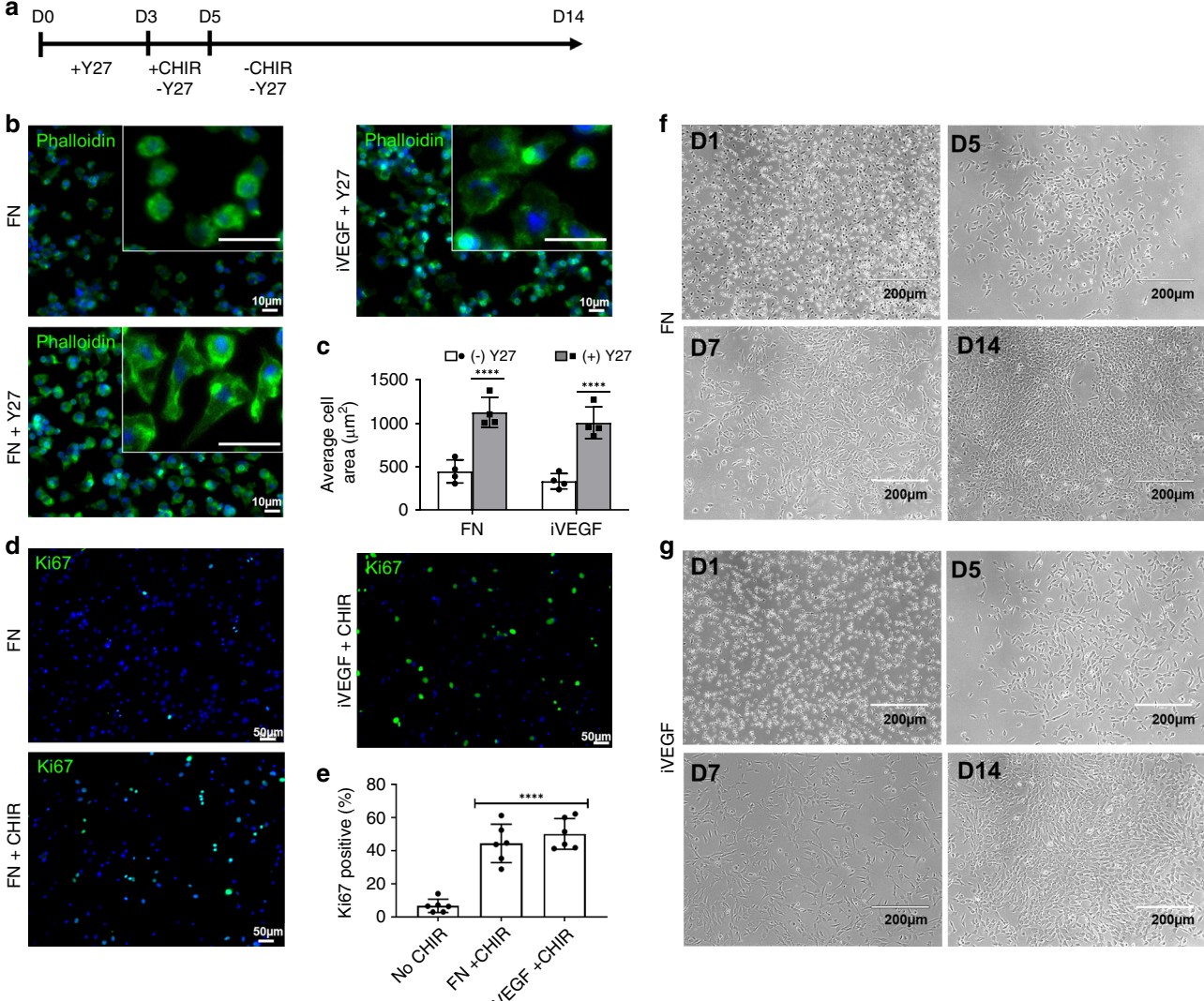

**Fig. 4 Optimized Protocol for MC Differentiation. a** Schematic of optimized protocol timeline. **b** Representative images of Y27 induced spreading of MC on both FN and iVEGFs and subsequent quantification of cell area (μm$^2$) (**c**) of phalloidin (green) stained cells. +Y27 (gray bars) and −Y27 (white bars); Significance denoted by (****$p < 0.0001$) compared to surfaces without Y27 using two-way ANOVA with Sidak's test (DF = 12, $F = 81.82$); $n = 4$ independent biological replicates; error bars denote ±SD of the mean. **d** Representative images of MC stained for the proliferation marker Ki67 (green) after CHIR treatment for 2 days and subsequent quantification of positive cells (**e**) four asterisk denotes statistical significance ($p < 0.0001$) compared to No CHIR treatment using one-way ANOVA and Dunnett's test (DF = 15, $F = 42.32$); $n > 300$ cells per $n = 6$ independent biological replicates; error bars denote ±SD of the mean. **f, g** Representative brightfield images of MC at the indicated times during the 14-day differentiation on FN (**f**) or iVEGF (**g**).

changed into a cobblestone morphology as cells proliferated and formed tight colonies typical of EC (Fig. 4f, g).

Over the 14-day differentiation, we assessed multiple genes involved in EC differentiation and function as well as genes associated with pro-inflammatory (M1) and anti-inflammatory (M2) macrophage activation states (Fig. 5). Interestingly, the differentiation process resulted in a genotype comprising both classic endothelial as well as macrophage genes, especially M2-associated genes. While some monocyte genes were quickly downregulated such as *CD14* (Fig. 5a) and *CX3CR1* (Supplementary Fig. 2A), the non-classical monocyte marker *CD16* was upregulated nearly 100-fold during initial differentiation and decreased to pre-differentiation levels by day 14 (Fig. 5b). A similar trend was observed for M1 associated genes such as *IL6, IL12, iNOS*, and *TNF-α* (Fig. 5c–e, Supplementary Fig. 2a). Conversely, some M2 associated genes were dramatically upregulated during differentiation and remained elevated above pre-differentiation levels such as *CD163*, a gene that is completely lacking in EC such

as HUVEC and human carotid endothelial cells (HCAEC) (Fig. 5f). Other M2-associated genes followed a similar trend including *EGR2, IL10, ARG1*, and *FN* (Fig. 4g, h, Supplementary Fig. 2a).

Several transcription factors that are well known to be crucial to EC differentiation during development were upregulated during differentiation. Interestingly, the pattern of upregulation loosely followed the accepted temporal sequence of EC transcription factors during development. *ETV2* and *GATA2* are transcription factors that appear early in EC development and as shown reach maximum upregulation on day 5 (Fig. 5i, j). Following the paradigm, *SOX17* reached maximum upregulation after *ETV2* and *GATA2*, on day 7 and remained elevated similar to typical EC levels at later times (Fig. 5k). Another important EC transcription factor, *ERG* was upregulated early during differentiation and then decreased but remained at levels similar to HUVEC (Fig. 5l). Interestingly, the transcription factor *HEY1*, a venous EC differentiation marker that is expressed in HUVEC but not HCAEC, was significantly downregulated during MC

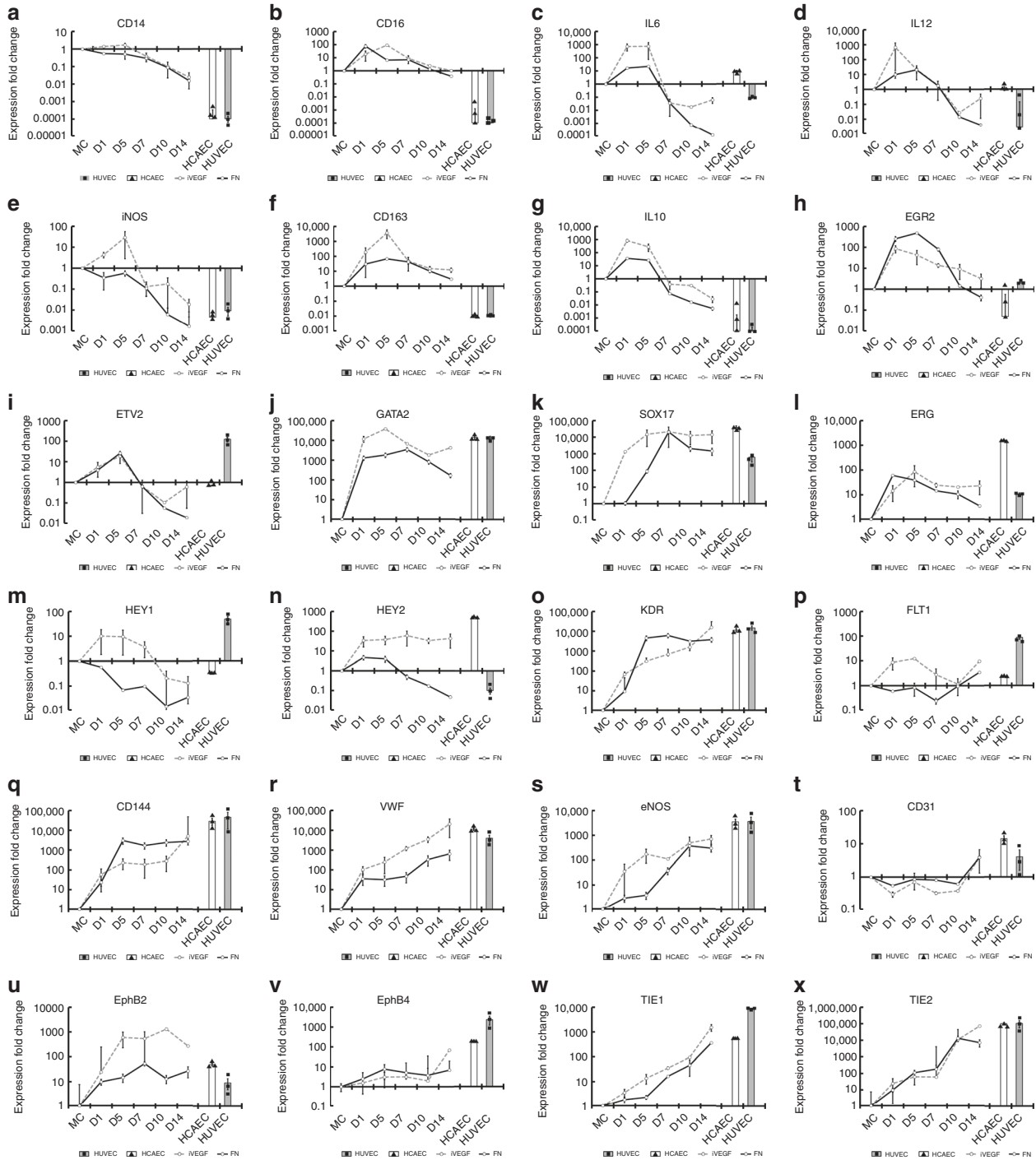

**Fig. 5 Gene expression profile as MC differentiate towards EC.** Quantification of gene expression via quantitative PCR over MC at day 0. All genes were internally normalized to RPL32 cycle number. HUVEC (gray bars) and HCAEC (white bars) gene expression served as a comparison references (**a–x**). Monocyte markers (**a**, **b**); M1 macrophage markers (**c–e**); M2 macrophage markers (**f–h**); Endothelial transcription factors (**i–n**); Endothelial functional markers (**o–x**). Error bars indicate ±SD of the mean over *n* = 3 independent biological replicates. Each PCR reaction was conducted with triplicate for each gene.

differentiation (Fig. 5m). Conversely, its arterial counterpart expressed in HCAEC but not in HUVEC, *HEY2*, was upregulated early on but later decreased on FN. Interestingly, *HEY2* was significantly upregulated by nearly 100-fold, and remained at high levels on iVEGF (Fig. 5n).

Next, we assessed markers of mature and functional EC. Of particular interest is *KDR/VEGFR2*, which was not expressed in MC

as demonstrated already using flow cytometry (Fig. 3b, c). As indicated in Fig. 5o, *KDR* was dramatically upregulated during differentiation, reaching EC levels by day 14. *FLT1/VEGFR1* expression was upregulated early on iVEGF but then returned to similar levels as FN before increasing to EC and MC levels (Fig. 5p). *VE-Cadherin/CD144* expression was upregulated significantly and remained at high levels through day 14 (Fig. 5q). Similarly, *vWF*,

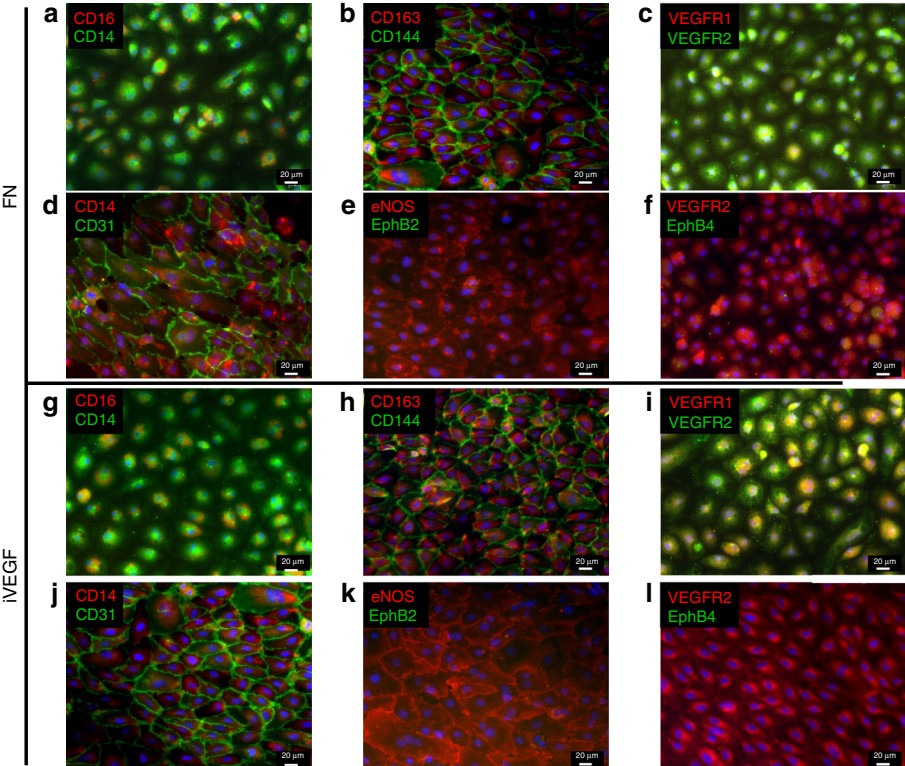

**Fig. 6 Phenotypic changes during MC-derived EC.** Immunostaining for MC and EC markers after 14 days of differentiation on FN (**a**–**f**) or iVEGF (**g**–**l**). **a**, **g** CD16 (red) and CD14 (green); **b**, **h** CD163 (red) and CD144 (green); **c**, **i** VEGFR1 (red) and VEGFR2 (green); **d**, **j** CD14 (red) and CD31 (green); **e**, **k** eNOS (red) and EphB2 (green); **f**, **l** VEGFR2 (red) and EphB4 (green). Scale bars: 20 μm. Representative images from n = 10 independent biological replicates.

which is expressed in mature EC, was upregulated and continued to increase to EC levels by day 14 (Fig. 5r). Furthermore, the functional EC marker, *eNOS* was also upregulated to EC levels, with iVEGF conferring higher expression of *eNOS* during early differentiation (Fig. 5s). MC/Mφ and EC both express *PECAM1/CD31*, therefore, it is not surprising that expression of *CD31* was fairly stable at MC levels with a final upregulation towards EC levels, which are approximately 5-fold higher than MC (Fig. 5t). Surprisingly, and similar to the shear related transcription factors *HEY1* and *HEY2*, the arterial EC marker *EphB2* was dramatically upregulated during differentiation on both FN and iVEGF, with iVEGF conferring much greater expression of *EphB2*, similar to the arterial HCAEC and above the venous HUVECs (Fig. 5u). Conversely, the venous EC marker *EphB4* was slightly upregulated during MC to EC differentiation but remained low compared to the expression levels observed in HUVECs or HCAECs (Fig. 5v). Other EC markers were also upregulated during differentiation such as *TIE1, TIE2, NRP1*, and *NRP2* as indicated in Fig. 5w, x and the heat map of Supplementary Fig. 2a, b. Similar results were obtained when cells were cultured on SIS substrate functionalized with heparin and VEGF (SHV) (Supplementary Fig. 2C).

**Monocytes differentiate to an EC and M2 macrophage mixed phenotype.** Next, we investigated the expression of endothelial proteins KDR, eNOS, CD31, and CD144 by immunocytochemistry (Fig. 6; Supplementary Fig. 3: secondary antibody assay controls and Supplementary Fig. 4: endothelial cell assay controls). Interestingly, the monocyte markers CD14 and CD16 were still present by day 14, albeit expression of CD16 was low (Fig. 6a, g). Interestingly, cells formed VE-cadherin (CD144) junctions, while maintaining expression of the M2 macrophage marker CD163 either on FN or iVEGF (Fig. 6b, h). Similarly, cell junctions contained CD31, while cells continued to express CD14

(Fig. 6d, j). Initially MC lacked expression of VEGFR2, however, VEGFR2 was highly expressed by day 14 of differentiation (Fig. 6c, i). Finally, Mc-derived EC expressed the phosphorylated form of eNOS, indicating acquisition of EC function but lacked expression of EphB2 or EphB4 at the protein level (Fig. 6e, f, k, l).

**MC-derived EC develop EC function.** To assess the function of MC-derived EC, we employed acetylated LDL (acLDL) uptake and neo-vessel (tube) formation in vitro. After 14 days of differentiation, cells were assessed for acLDL uptake and the percentage of acLDL+ cells quantified as shown in Fig. 7a. Of note, only 6.4 ± 3.3% (n = 3 independent biological assessments) of MC could uptake acLDL. In comparison, when MC were activated to a traditional macrophage phenotype, the percentage of cells that could uptake acLDL increased to 67 ± 9.3%. When MC were differentiated towards EC either on FN or iVEGF using our defined protocol, 92 ± 5.6% and 88 ± 3.9% of cells could uptake acLDL, similar to HUVEC and HCAEC (85.3 ± 7.2% and 93.6 ± 8.4%, p < 0.0001 as compared to initial MC).

Although both macrophages and EC uptake acLDL, only EC are known to organize into neo-vessels/tubes on matrigel. Interestingly, MC-derived EC on FN or iVEGF formed tubes within 24 h of adhesion to matrigel. In addition, cells within the neo-vessels expressed VE-cadherin/CD144 and CD31 that were localized at the cell junctions. Surprisingly, these cells also expressed CD16 as indicated in Fig. 7b, c.

**Shear stress augments MC-EC differentiation and EC function.** Shear stress is present at all times after implantation in vivo and is well known to induce EC differentiation into a mature phenotype and alignment along the direction of flow. To examine the effect of shear stress on MC to EC differentiation, MC cells were differentiated for 9 days under static conditions; then subjected to

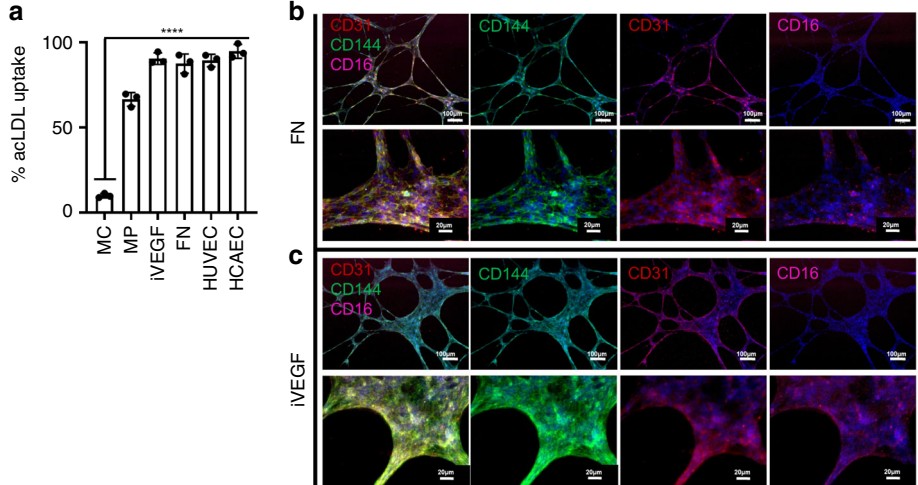

**Fig. 7 MC-derived EC develop endothelial function. a** MCEC differentiated on FN or iVEGF uptake acetylated LDL. MC on day 0 are unable to uptake LDL compared to traditionally activated, LPS stimulated MΦ, MCEC on FN or VEGF or HUVEC or HCAEC. (****$p < 0.0001$) statistical significance using one-way ANOVA and Sidak's test (DF = 12, $F = 205.1$); $n = 3$ independent biological replicates; error bars indicate ±SD of the mean. MCEC differentiated on FN (**b**) or VEGF (**c**) formed tubular networks when cultured on Matrigel. Representative immunostaining for the monocyte markers CD16 (pink), endothelial marker CD144 (green), and CD31 (red) over $n = 5$ independent biological replicates.

gradually increasing shear stress by ramping from low to high shear stress (1 to 10 dyn/cm²) over 2 days; and further cultured under high shear stress for 3 days (for a total of 14 days of differentiation). As shear stress is known to induce NO production, we quantified $NO_2$ content via Griess reagent. Indeed, shear induced $NO_2$ production significantly, as compared to static conditions when cultured on FN and especially on iVEGF at levels similar to HUVEC and HCAEC (Fig. 8a).

Next, we assessed the effect of high shear on a number of key EC genes that are known to be affected by shear stress (Fig. 8b–k). Consistent with the increased NO production, *eNOS* was upregulated dramatically with shear (Fig. 8b). *CD144/VE-cadherin* was similarly upregulated by shear, with expression matching EC levels (Fig. 8c), as did key EC mechanosensory proteins such as *JAG1*, *DLL4*, and the receptors *NOTCH1* and *NOTCH4* (Fig. 8d–g). Interestingly, high shear downregulated the venous genes *HEY1* and *EphB4* and upregulated arterial EC genes *HEY2* and *EphB2* (Fig. 8h–k).

Following gene expression changes, we assessed protein expression upon exposure to shear. Under high shear stress conditions, the cells aligned parallel to the direction of flow as expected (Fig. 8l–s), with CD144 clearly delineating the cell borders (Fig. 8l, p). Immunostaining showed that both the MC marker CD14 and the M2-MΦ marker CD163 remained consistently expressed on FN and iVEGF (Fig. 8l, m, p, q). Notably, high shear induced expression of EphB2 - but not EphB4 - at the cellular junctions, typical of mature arterial endothelial cells (Fig. 8n, o, r, s). On the other hand, when cells were exposed to low shear (1 dyne/cm²) for 5 days, cell alignment was not as evident, and expression of arterial EC marker EphB2 was not apparent (Supplementary Fig. 5a, c), but the venous EC marker EphB4 was weakly expressed (Supplementary Fig. 5b, d), indicating that shear plays an important role in the acquisition of MC-EC phenotype, similar to bona fide EC.

**MCEC are distinct from MC and cluster with mature EC.** Furthermore, using single cell RNA-sequencing (scRNA-Seq) we examined the global transcriptome of individual cells in the three cell populations: starting MC pooled from three donors after 1 h adherence to FN; MC-derived EC (from the same three donors) subjected to shear stress; and HCAEC subjected to shear stress.

Uniform Manifold Approximation and Projection (UMAP) dimensionality reduction was applied to the integrated dataset and showed that the initial population of MC clustered separately from HCAEC, while the resulting MCEC overlapped with HCAEC and not MC (Fig. 9a). Co-expression analysis reveals that only the resulting MCEC contain cells co-expressing *KDR* and *PROM1/CD133*, whereas our starting MC population contained only one cell expressing *KDR* and a separate cell expressing *PROM1* (Fig. 9b), suggesting that our initial MC population did not contain any EPC/ECFCs, which are known to co-express *KDR/VEGFR2* and *PROM1/CD133*. Interestingly, heatmaps constructed using EC and MC associated genes revealed that the resulting MCEC cells expressed predominantly EC genes but also some MC genes (Fig. 9c), in agreement with the RT-PCR data (Fig. 5). Violin plots of individual gene expression in each population further support this point (Supplementary Fig. 6).

**MCEC are derived exclusively from MC and not contaminating EC.** To provide further evidence of the differentiation potential of MC into EC and exclude the possibility that MCEC originated from rare circulating EPC/EC in the initial MC population, we employed a MC-specific promoter to select the initial MC population before the onset of differentiation. Specifically, we used a lentiviral vector (pCD68-ZsG-Puro) encoding for ZsGreen and Puromycin phosphotransferase under the control of the MC specific promoter, CD68[34,35] (see schematic in Fig. 10a). After transduction of MC with pCD68-ZsG-Puro and puromycin selection, all cells were ZsGreen+ (Fig. 10b), indicating active CD68-Pr. As a control, HCAEC cells were also transduced with the same vector but all cells died upon puromycin selection, suggesting that CD68-Pr was not active in EC as expected.

Additionally, HCAEC were transduced with a dual promoter lentiviral vector (pCD68-LVDP) encoding for ZsGreen under the CD68-Pr and DsRed2 under the human (h)PGK promoter (Fig. 10a). This vector was developed in our laboratory and contains insulator and terminator sequences that diminish promoter interference[36,37]. Transduced HCAEC with pCD68-LVDP expressed DsRed but not ZsGreen (Fig. 10c), in agreement with the lack of CD68-Pr activity seen with puromycin selection.

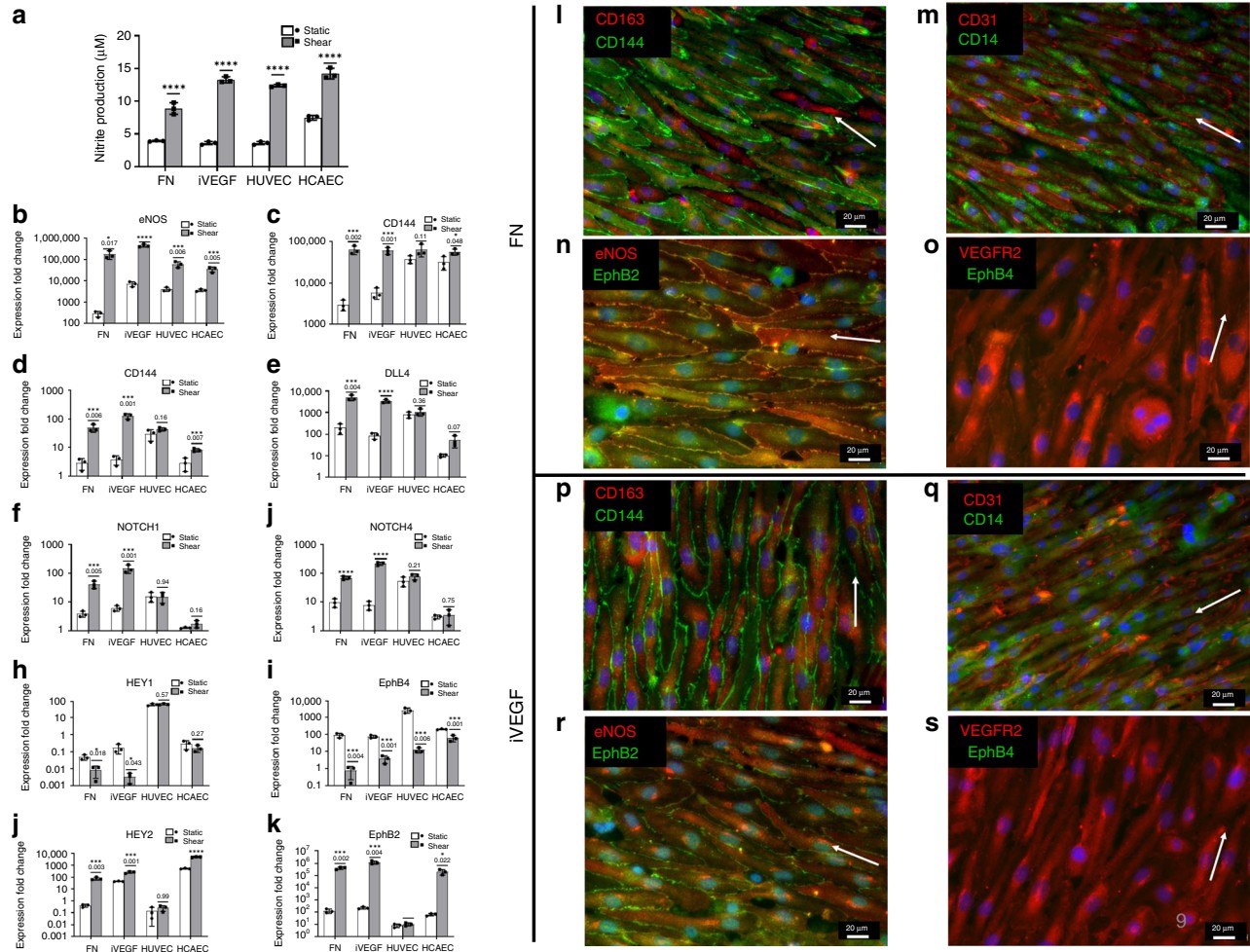

**Fig. 8 Shear augments MC-to-EC differentiation. a** High shear (10 dynes/cm$^2$) induces production of NO production in MC-EC cells differentiated on FN or iVEGFs as compared to static condition (gray bars). HUVEC and HCAEC were used as reference ECs capable of producing NO under shear. Statistical significance as indicated by p-values (****p < 0.0001) determined by two-way ANOVA and Sidak's test (DF = 16, F = 1390), n = 3 independent biological replicates; error bars denote ±SD of the mean. **b-k** MC-EC were subjected to high shear (10 dynes/cm$^2$) and expression of the indicated genes was measured by RT-PCR. Genes were internally normalized to RPL32 and compared to MC at day 0. Static (white bars) and shear (gray bars). HUVEC and HCAEC served as a reference EC cells. Statistical significance as indicated by p-values (****p < 0.0001) determined by two-way ANOVA and Sidak's test (DF = 16, F = 1390), n = 3 independent biological replicates; error bars denote ±SD of the mean. **l-s** Immunostaining for key MC and EC markers after exposure to high shear. FN or iVEGFs: (**l, p**) CD163 (red) and CD144 (green); (**m, q**) CD31 (red) and CD14 (green); (**n, r**) eNOS (red) and EphB2 (green); (**o, s**) VEGFR2 (red) and EphB4 (green). White arrows indicate the direction of fluid flow. Scale bar: 20 µm. Representative images of n = 5 independent biological replicates.

These results show that CD68-Pr is active in MC but not in EC and therefore, can be used to track MC as they differentiate.

Selected MC were subjected to the MCEC differentiation protocol as discussed above. Interestingly, after 14 days of differentiation, the selected cells maintained ZsGreen expression and continued to express the MC marker, CD14. At the same time, these cells also expressed the EC-specific proteins, CD144 and VEGFR2 (Fig. 10d). Notably, upon application of shear stress, expression of both ZsGreen (CD68-Pr activity) and CD14 decreased, while EC markers CD144 and VEGFR2 remained highly expressed in all cells (Fig. 10e), demonstrating that MC, and not a contaminating EC fraction, could differentiate into EC and that shear stress contributed significantly to the MCEC phenotype.

## Discussion

Various strategies have been employed to capture rare circulating endothelial progenitor cells, EPCs, from the blood to induce endothelialization of implanted materials[28,31,38–40]. Some groups

employed antibodies against EPC specific proteins such as VEGFR2, CD34, and CD133, but application in small animal models was met with varying success[40,41]. In addition, the use of antibodies was shown to reduce functionality of the cell surface protein, especially in the case of VEGFR2[40]. SDF1α, a cytokine that binds to CXCR4 on EPCs[27,28,38], has also been employed but resulted in incomplete endothelialization in the middle of the grafts and neo-intimal hyperplasia[28]. In recent work from our laboratory, we employed VEGF to capture circulating cells that express the VEGF receptor and endothelialize the otherwise acellular grafts[30,31]. This approach was very successful as shown by high patency rates (92%) and successful remodeling of arterial grafts in both a mouse model and the clinically relevant ovine model. Even though our in vitro studies indicate small differences between iVEGF and FN culture surfaces, likely due to the presence of soluble VEGF in both conditions, in vivo VEGF was essential for maintaining patency and promoting remodeling. In contrast, FN based grafts have been previously shown to fail[27], likely due to the RGD integrin binding domain present in FN that

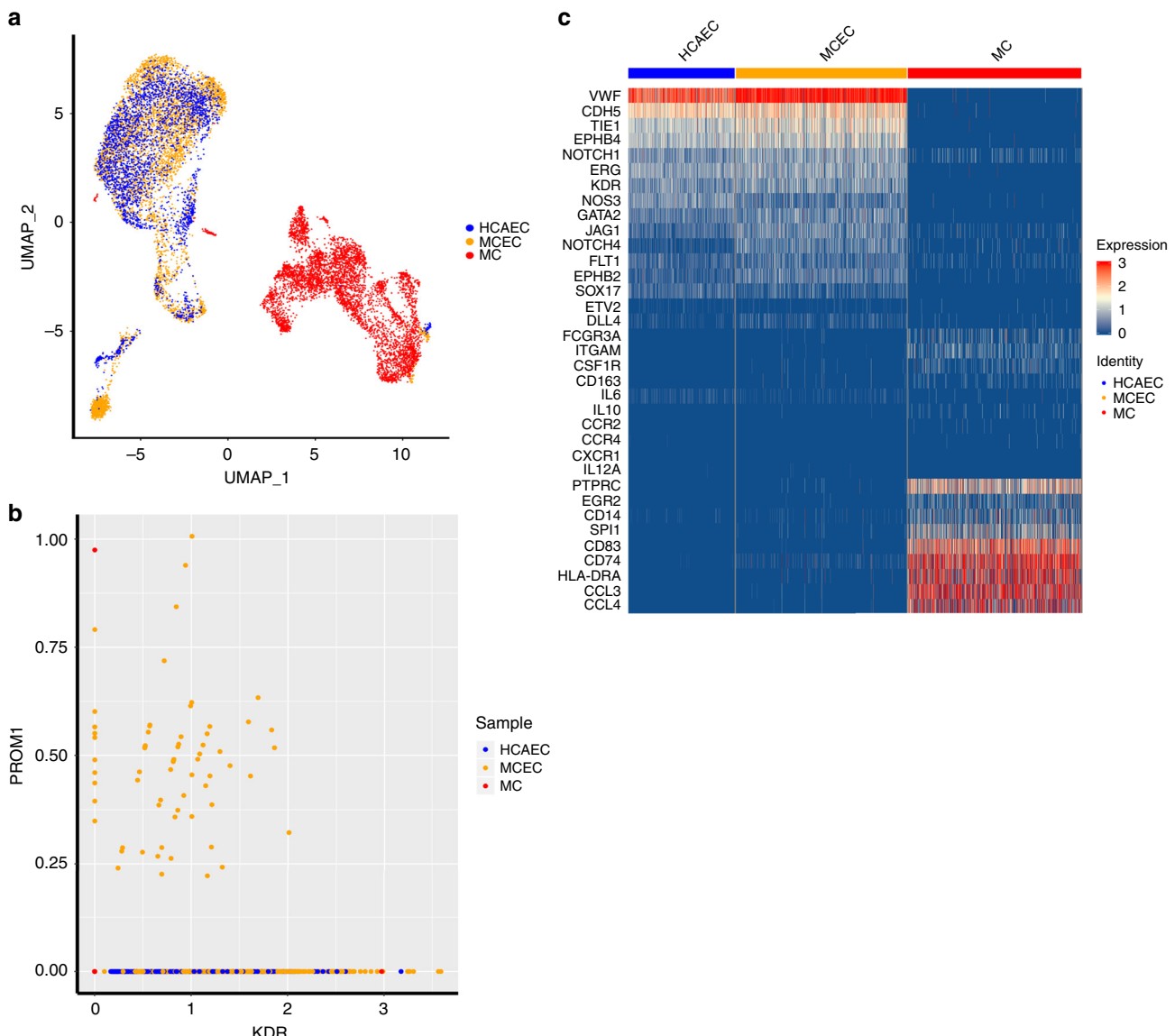

**Fig. 9 Single-cell RNA sequencing.** Single-cell RNA sequencing of three cell populations, initial MC, MCEC, and HCAEC both subjected to shear stress. **a** UMAP reduction analysis of unsupervised data of all three cell populations. **b** Co-expression analysis of KDR and PROM1 expression across all three populations. MC do not express KDR or PROM1. **c** Heat map representation of MC, M1, M2, and EC genes across all three populations.

binds a plethora of cell types, including platelets[42–45]. In the absence of VEGF, grafts failed due to occlusion within hours of implantation in the ovine model, suggesting that heparin alone was not enough to prevent clotting. However, in mouse models, grafts without VEGF did not fail; likely due to the significantly higher flow rate in mouse abdominal aorta compared to the carotid artery of sheep. In addition, heparin only grafts exhibited a high degree of inflammation and lacked a proper endothelium and medial layer. In contrast, VEGF grafts were fully endothelialized within 1 month, consisted of pro-regenerative-anti-inflammatory macrophages and exhibited distinct vascular remodeling towards the native state[29]. Therefore, VEGF promotes capture of cells with patency inducing, pro-regenerative capacity in vivo.

In this study we show that the cells populating the lumen of VEGF-decorated grafts were mostly VEGFR1-expressing MC. Interestingly, VEGFR1 has higher affinity for the VEGF ligand than VEGFR2[46], which is expressed in EC, and therefore may enable higher selectivity of the VEGF surface towards MC. In

addition, MC outnumber EPCs to a great extent, as EPCs represent <0.01%, whereas MC represent over 20% of PBMNC[47]. Indeed, VEGF grafts did not contain any EC on the lumen at 1-week post implantation and instead, the graft lumen was covered in MC-M2 polarized cells (CD14+/CD163+). Binding of MC to immobilized VEGF was also shown in vitro using microfluidic channels with immobilized VEGF that captured MC cells from whole blood under flow. While we cannot exclude the possibility that some EPCs might have been captured on the graft lumen, the vast majority of cells on the graft lumen continued to express MC-Mφ markers at 1 month and even at 3 months post-implantation, further supporting the hypothesis that they originated from MC.

MC are known to play a crucial role in angiogenesis and arteriogenesis. During angiogenesis, Mφ have been found to orchestrate bud development of new vessels and are often found in direct contact with tip cells[48]. Similarly, MC-derived, tumor associated Mφ, are known to induce rapid angiogenesis around growing tumors and have been shown to be present within the

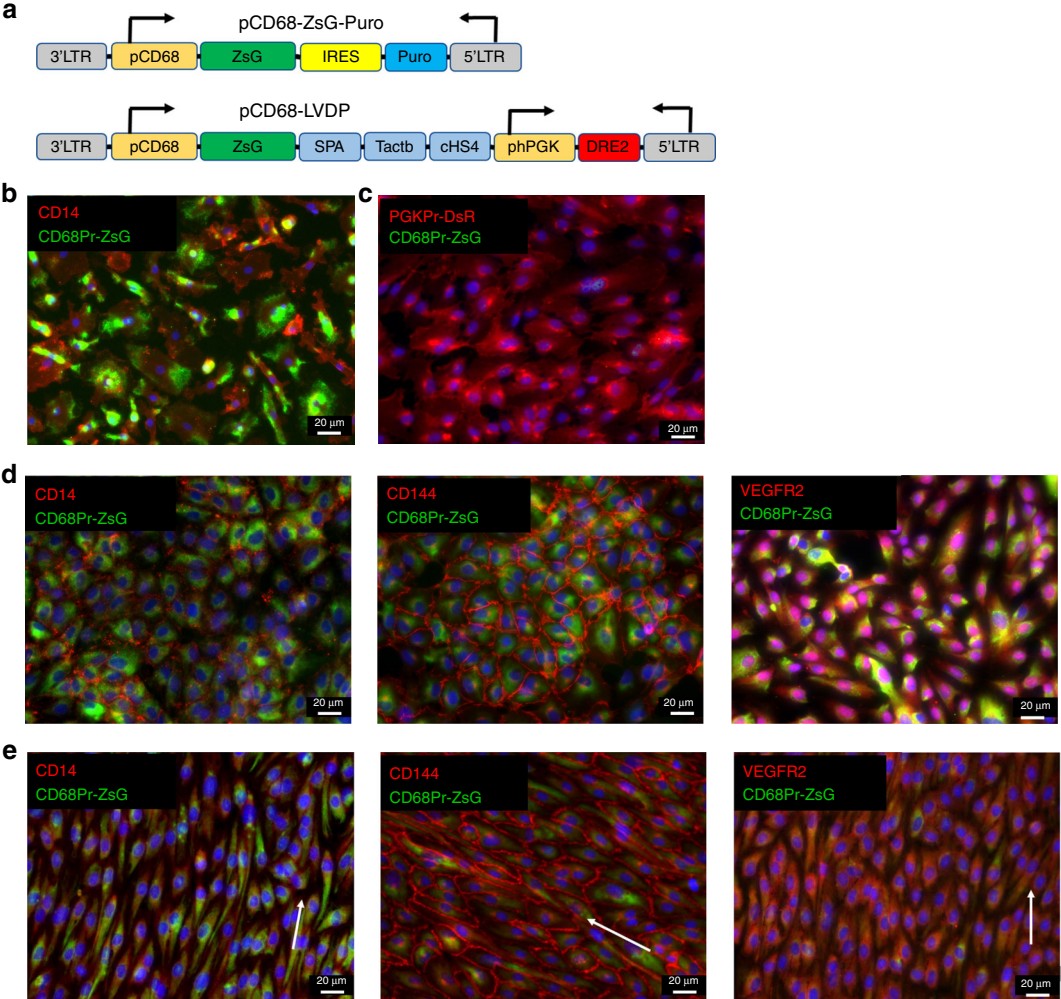

**Fig. 10 CD68 promoter driven selection of MC for differentiation. a** Schematic of lentiviral vectors encoding for ZsGreen and puromycin phosphotransferase under the CD68 promoter (CD68-Pr; pCD68-ZsG-Puro) and Schematic of lentiviral dual promoter (LVDP) encoding for ZsGreen under the CD68-Pr and DsRed under the human (h)PGK promoter (pCD68-LVDP). The two transcriptional units are separated by polyadenylation (SPA, synthetic polyA), terminator (Tactb), and insulator (cHS4) sequences that diminish promoter interference[36,37]. **b** Immunostaining of MC transduced with lentiviral vector shown in (**a**) and selected with puromycin (ZsGreen+) for the MC marker CD14 (red) prior to differentiation. **c** HCAEC cells were transduced with the lentiviral dual promoter vector shown in **b**. They express DsRed but not ZsGreen indicating that CD68 is inactive in HCAEC. **d**, **e** Immunostaining for MC marker CD14 and EC markers CD144 and VEGFR2 (red) of puromycin selected MC (ZsGreen+) that were coaxed to differentiate towards EC under (**d**) static or (**e**) shear stress conditions. Scale Bar: 20 μm.

growing vessels in contact with tip cells[49]. MCs were also shown to be critical for the success of implanted grafts[50,51], as ablating them using genetic or chemical means led to graft failure due to stenosis and lack of proper endothelium[50]. In addition, bone marrow myeloid progenitor cells were shown to induce re-endothelialization after balloon injury in rats[52]. All these studies show that MC are critical for angiogenesis and endothelialization, but their contribution is thought to be indirect through secretion of growth factors and cytokines that promote endothelial proliferation and migration.

In contrast to these studies, here, we report for the first time, direct incorporation of MC-derived ECs into the endothelium of a neo-artery in a large animal model. At 1-week post-implantation, acellular grafts were coated with cells expressing MC but not EC proteins. At 1 and 3 months post-implantation, the luminal cells retained MC/Mφ markers while also expressing functional EC markers, including expression of eNOS and production of NO, ultimately conferring graft patency. Interestingly, similar results were obtained in our mouse model, whereby the VEGF grafts developed a well-defined endothelial layer comprised of cells

expressing both EC and MC/Mφ markers[29]. In contrast to small animals, where ECs can migrate from the anastomotic sites[53,54], in humans and large animals such as sheep, EC ingrowth does not occur[54,55]. Therefore, endothelialization by an abundant cell in the blood such as MC may be critical for the success of arterial grafts and perhaps also venous or cardiac transplants. This novel mechanism may also contribute to the repair of cardiovascular tissues in-vivo following EC disruption by injury or disease.

Previous studies have shown that MC can express EC genes after in vitro culture[56–61] but lack VE-cadherin positive adherens junctions and the ability to form tubes, indicating lack of EC functionality. In contrast, we developed a strategy to coax MC to differentiate into functional EC like cells in vitro. Our strategy involved several steps. First, we utilized the ROCK inhibitor, Y27, which was used in previous studies to promote spreading and prevent death of embryonic stem cells during sub-culture[62]. We also substituted serum with PRP, which has been used previously to enhance the survival of MC[63]. Most important, activation of the *WNT* pathway via CHIR was necessary to coax MC towards a functional EC phenotype, exhibiting VE-cadherin junctions and

tube formation on matrigel. The *WNT* pathway is known to be activated under physiological shear conditions[64], suggesting that WNT activation might have simulated those conditions in vitro. MC and EC share a common developmental origin stemming from the hemogenic endothelium and *WNT* is indispensable for EC development[65,66]. In this context, the requirement for *WNT* activation during the MC-to-EC differentiation may not be completely unexpected. Indeed, we found that application of shear stress dramatically improved the expression of EC genes, induced cell alignment in the direction of flow and formation of VE-Cadherin adherens junctions. Most notably, shear stress increased expression of the key arterial marker *EphB2* while downregulating the venous marker *EphB4*, clearly suggesting that MC-derived EC could develop arterial phenotype in a high shear stress environment. These results demonstrate that MC could turn into functional arterial EC under conditions that mimic the arterial environment (PRP, high shear stress), thereby providing strong support of the in vivo data.

We have taken steps to ensure that MCEC originated from MC instead of the rare EPC population in blood. First, cells were captured on FN or iVEGF for a short time (1 h) and unbound cells were washed away before initiating differentiation. In contrast, in protocols isolating EPCs from blood, this initial attachment serves to remove monocytes/macrophages while unbound cells are re-plated on ECM coated dishes for additional time to allow for EPC attachment to occur. Furthermore, immunostaining showed that even after 14 days of differentiation cells stained positive for CD14/CD16/CD163, all MC markers that are not present on EC or EPC cells. Third, scRNA-seq clearly showed that the initial MC population was distinct from HCAEC and contained no cells co-expressing *KDR/VEGFR2* and *PROM1/CD133*, which are known to be exclusive to EPC/ECFC. In contrast, the resulting MCEC population was significantly similar to mature HCAEC, while also expressing some MC/MΦ genes, in agreement with our RT-PCR results. Finally, we employed a MC/MΦ specific CD68 promoter[34,35] to select for MC by removing any other cells that might be present, including EC. Indeed, after selection all cells expressed CD14 and after 14 days of differentiation these cells expressed mature EC proteins, such as CD144 and VEGFR2, while maintaining expression of CD14. Collectively, these data suggest that MCEC originated from blood MC and not from any rare EPC contaminating the MC population.

In light of our findings, it is plausible that MC may be even more plastic than originally thought, depending upon the microenvironment they are in. The literature suggests a multitude of macrophage identities under the "M2 umbrella", with M2a, M2b, M2c, and M2d already well established[67–69]. Our study suggests that there may be a fifth M2 type, M2-endothelial cell or M2e that can turn into an EC when the need for endothelialization arises e.g. in response to injury, disease or implants. Activation of M2e may depend on signals such as VEGF that recruit MC and coax them towards an EC fate.

In this study we report, for the first time, a novel mechanism of endothelialization of vascular grafts through incorporation of monocyte derived endothelial macrophages, M2e, in a large animal model. In addition, we establish a protocol to generate M2e cells through activation of the WNT pathway and application of shear stress. While rare EPCs are known to be involved in endothelium repair, our study indicates that the highly prevalent circulating monocytes may provide a novel, direct, and efficient strategy of reendothelializing acellular arterial grafts decorated with VEGF and perhaps other ligands as well.

## Methods

**VEGF cloning and protein production.** Bioactive, recombinant VEGF isoform 165 was produced and purified as previously described[31]. Briefly, bacteria strain

*Escherichia coli* BL21-DE3-pLysis containing pGEX-VEGF, encoding for a thrombin cleavable glutathione-s-transferase (GST) tag followed by the *VEGF-165* gene, was induced with 1 mM isopropyl β-D-1-thiogalactopyranoside for protein production for 4–6 h at 37 °C and 300 rpm. Bacterial pellets were then lysed (50 mM Tris, 500 mM NaCl, 1 mM ethylenediaminetetraacetic acid (EDTA), pH 8.5, 1 mg/mL lysozyme, and protease inhibitors) and sonicated. GST-VEGF-containing inclusion bodies were then subjected to numerous rounds of washing and sonication then solubilized (50 mM Tris, 500 mM NaCl, 7 M Urea, 1 M Guanidine-HCl, 1 mM EDTA, 100 mM dithiothreitol (DTT), pH 8.5) prior to refolding by dialysis. Briefly, solubilized GST-VEGF was immediately added to a dialysis membrane (SpectraPor-1 6–8 kDa cut-off) and dialyzed in 100× volume of Refolding Buffer-1 (50 mM Tris, 500 mM NaCl, 10 mM KCl, 1 mM EDTA, 2 M Urea, 500 mM L-Arginine, 5 mM reduced glutathione, 0.5 mM oxidized glutathione, pH 8.5) for 24 h, then the refolding buffer was replaced with half the urea concentration of the previous day for 72 h. The final dialysis step was performed in PBS. GST was cleaved from VEGF using thrombin (100 units/mg protein; Sigma-Aldrich; St. Louis, MO). Refolded VEGF was then subjected to purification using Hitrap Heparin Column (GE Healthcare; Chicago, IL) according to the manufacturer's instructions.

**Cell culture.** HUVECs were purchased from Lonza (Allendale, NJ) as a pooled donor isolation, maintained below 75% confluence in EGM2 complete medium (Lonza), and used between passage 2 and 6. Human Carotid Artery Endothelial Cells (HCAEC) were purchased from Cell Biologics (Chicago, IL) as a pooled donor isolation, maintained below 75% confluence in EGM2 complete medium and used between passage 3 and 6. Hair follicle-derived mesenchymal stem cells (HF-MSC) were isolated as described and maintained in DMEM (Thermo Fisher Scientific; Grand Island, NY) supplemented with 10% FBS (Atlanta Biologicals; Atlanta, GA) and 1 ng/ml bFGF[70]. NIH-3T3 fibroblasts were purchased from American Type Culture Collection (ATCC) and maintained in DMEM supplemented with 10% FBS (Thermo Fisher). Ovine pulmonary artery endothelial cells (OPAECs) were isolated as previously described[71] and were maintained in DMEM supplemented with 20% FBS. Human dermal fibroblasts (h-dFB) were isolated as described previously from neonatal foreskin and maintained in DMEM supplemented with 10% FBS[72]. All media were supplemented with 1% antibiotic-antimycotic cocktail (Thermo Fisher). All cells were maintained in a humidified incubator with 5% CO₂ at 37 °C.

**Acquisition of monocytes from human whole blood and platelet apheresis cones.** Human whole blood was acquired from healthy donors as per University at Buffalo IRB/IACUC guidelines and federal regulations. Whole blood was acquired using traditional methods of needle and syringe, and supplemented with a tenth of acquired volume with sodium citrate and used within 4 h. To obtain platelet rich plasma (PRP), whole blood was acquired as described and centrifuged at 1500 rpm for 10 min to separate the RBC/WBC from PRP. PRP was carefully removed from above the RBC/WBC layer and frozen until use. Platelet apheresis cones, as a source of enriched WBC, were purchased from Roswell Park Comprehensive Cancer Center (Buffalo, NY). Cones were obtained from anonymous healthy donors. Cones were purged with PBS to remove the WBC rich blood and layered on top of histopaque-1077 (Sigma-Millipore; St. Louis, MO) to separate the WBC rich buffy coat from RBC. The buffy coat was then washed with PBS before use in experiments.

**Monocyte culture.** PBMNCs were first plated on fibronectin (FN; Thermo Fisher) or IVEGF for 1 h in EBM with no serum or PRP (basal media without supplements). Adherent cells were washed once in warm PBS to remove unbound cells. Adherent cells were further cultured up to 14 days in EGM2 medium, with all supplements except that FBS was replaced by autologous PRP (20% v/v). VEGF (50 ng/ml) and MCSF (1 ng/ml; Thermo Fisher) were added to the EGM2 medium throughout the culture period. On day 0, Y-27632, 10 μM, (Y27; Sigma) was added to the medium and removed on Day 3. On day 3, CHIR-99021, 10 μM, (CHIR, Sigma) was added to the medium and removed on day 5. EGM2 with 20% PRP, 50 ng/ml VEGF, and 1 ng/mL MCSF was replaced every 2–3 days.

**Culture under shear conditions.** On day 14, differentiated MCs on FN or iVEGF were further cultured in a bio-reactor setup as previously reported[73]. In this procedure circular dishes were modified with attachment of a smaller inverted circular culture dish inside of a larger dish. When this dish is placed on an orbital shaker, the force directs the medium to "swirl" around the center dish. Shear strength is then changed by changing the rotation speed. We adapted this procedure with an orbital shaker with a 1.9-cm radius. Shear was determined by the equation:

$$\tau = r\sqrt{\eta\rho(2\pi f)^3}$$

Where $\tau$ is the desired shear (dynes/cm²), $r$ is the radius of the orbital shaker, $\eta$ is the viscosity (poise), $\rho$ is the density (g/mL), and $f$ is the rotations speed in rounds per second. Using this equation and converting to rotations per minute, ~29, 84, and 133 rotations per minute equated to 1, 5, and 10 dynes/cm² of shear, respectively. Shear was slowly ramped up from 1 to 10 dynes/cm² over 2 days and

maintained at that level for 3 days prior to analysis. Medium was replaced every other day during shear studies.

**Immobilization of VEGF.** Chitosan (Sigma) was used at the manufacturer's recommended concentration of 0.1 mg/mL to coat tissue culture treated polystyrene surfaces (TC). Coating was performed by adding sterile chitosan solution for 12 h with rocking at 37 °C. After coating with chitosan, the surface was washed repeatedly with sterile water to remove unbound chitosan. Heparin (17–19 kDa) from porcine submucosa (Sigma) was dissolved in sterile water at a concentration of 5 mg/mL and then applied directly to the chitosan treated surfaces overnight at RT. Then the surface was washed with sterile water to remove unbound heparin and heparin binding to chitosan was determined using the toluidine blue binding assay as described previously[74].

Finally, recombinant VEGF, 10 μg/mL in phosphate buffered saline (PBS), was added to the Chitosan-Heparin surface. Binding of VEGF was optimal at 37 °C without rocking, for 2 h. VEGF binding was determined by ELISA using biotin-conjugated goat anti-VEGF antibody (100 ng/mL, 2 h, RT, R&D Systems), followed by incubation with streptavidin-HRP (1:200, 30 min, RT) and addition of substrate (TMB; Sigma). Absorbance was read at 450 nm using a Biotek Synergy 4 Spectrophotometer (with subtraction of background absorbance of 570 nm).

**Proliferation on VEGF functionalized surface.** To determine cell proliferation, HUVECs (under 70% confluent) were detached from tissue culture plates by treatment with 5 mM EDTA (10 min), re-suspended in basal EBM2 medium with 10% serum and plated at $5 \times 10^3$ cells per well on Chitosan-Heparin-VEGF, termed immobilized (i)VEGF, coated 48-well plates with varying concentrations of VEGF (as indicated in Fig. 1c). The cells were allowed to bind for 6 h and then unbound cells were removed and the medium was changed to EBM supplemented with 2% heat inactivated FBS. After 120 h, cells were subjected to MTT assay as described above.

**Capture of cells under flow in a microfluidic device.** Capture of endothelial cells under flow was assessed in a microfluidic device (Fig. 1a). The PDMS based device with channel dimensions of 400 μm in width, 200 μm in height, and 1 cm in length, contains four circular ports used for vacuum sealing to a flat surface. Functionalization of the surface with chitosan and heparin was performed as described above followed by addition of 50 μg/mL VEGF to obtain complete saturation of the CH surface in the microchannel. The PDMS based device was washed vigorously with 100% ethanol and dried under a stream of air before it was placed onto the functionalized surface. The input port was connected to a reservoir for medium and cells, and the output port was connected to a Harvard Apparatus Syringe Pump through a glass syringe (1 mL). The pump controlled the flow rate and therefore, the shear stress at the bottom surface of the micro channel, $\tau_w$, in the device could be calculated according to the equation:

$$\tau_w = \frac{6\mu Q}{h^2 w_1}$$

where, $\mu$ is the viscosity; $Q$ the volumetric flow rate; $h$, the height; and $w_1$, the width of the micro channel. DMEM without serum was used during fluidic runs with a viscosity of 0.88 cP[75]. All cells (under 70% confluent) were detached from tissue culture plates by treatment with 5 mM EDTA (10 min) and re-suspended in DMEM medium without serum at the required concentration prior to running on the device. When indicated, cells were pre-stained with fluorescent carbocyanines, DIO or DIL (Thermo Fisher) according to the manufacturer's specifications to enable live cell tracking.

**Flow cytometry.** Flow cytometry was performed on three different MC isolation methods, buffy coat PBMNCs, fibronectin (FN) captured PBMNCs, and iVEGF captured PBMNCs. FN surfaces were prepared using human FN (Thermo Fisher) at 10 μg/mL in PBS overnight at 4 °C . Surfaces with VEGF were prepared as just discussed. For FN and iVEGF, buffy coat PBMNCs were allowed to adhere to either FN or iVEGF for 1 h at 37 °C /5% CO₂ in the absence of serum; gently washed with PBS to remove unbound cells; and then gently mechanically removed from the surface and fixed with 4% paraformaldehyde. Fixed cells were blocked in 5% goat serum in PBS, washed, and incubated for 1 h on ice with primary conjugated antibodies. The following fluorescent conjugated antibodies were used to characterize isolated cells, CD14-FITC, CD16-BV421, VEGFR2-Alexa Fluor-647 (BD Biosciences; Franklin Lakes, NJ) and VEGFR1-PE (Miltenyi Biotech; Bergisch Gladbach, Germany). IgG1 isotype controls for each conjugated dye was run to establish proper gating. Flow cytometry was performed using a BD Fortessa X-20 (BD Biosciences) and data analyzed with FCS Express software suite (DeNovo Software; Naples, CA).

**Cloning and lentivirus production and transduction.** The CD68 promoter sequence in the pcDNA3-CD68 vectors was a gift from Peter Murray (Addgene plasmid #34837; http://n2t.net/addgene:34837;RRID:Addgene_34837). The minimal CD68 promoter sequence was amplified with primers containing restriction sites (Age1 and Nhe1) and sub-cloned upstream of ZsGreen into a self-inactivating lentiviral vector containing the sequence ZsGreen-IRES-Puro to enable selection via puromycin and cell tracking via ZsGreen (pCD68-ZsG-Puro). Furthermore, a

dual promoter vector containing the CD68 promoter driving ZsGreen and human PGK promoter driving DsRed was produced via subcloning the CD68 promoter with restriction sites Bsth1 and Nhe1 upstream of the ZsGreen gene (pCD68-LVDP). Complete LVDP vector was developed within our laboratory and contains insulating sequences and was produced as described[37].

For lentivirus production, 293T/17 cells were transfected with three plasmids (16.8 μg lentiviral vector, 15 μg psPAX2, and 5 μg pMD.G), using the standard calcium phosphate precipitation method. Virus was harvested 24 h post transfection, filtered through 0.45 μm filter (Millipore, Bedford, MA, USA), pelleted by ultracentrifugation (50,000×g at 4 °C for 2 h) and resuspended in fresh medium. Cells were transduced in the presence of 8 μg/ml polybrene for 4 h, washed, then provided fresh medium. Cells recovered and began expressing ZsGreen in 48 h, then selected with puromycin (1 μg/mL) for 3 days and following selection they were subjected to the differentiation protocol as reported above.

**Gene expression analysis.** MCs were assessed by qRT-PCR for gene changes prior to differentiation (day 0), and at 24 h, 5 days, 7 days, 10 days, and 14 days after induction to differentiation. To evaluate the effect of shear stress, MCs were induced to differentiate for 9 days and then subjected to shear for 5 days (days 10–14) in the same differentiation medium. RNA was isolated and purified according to the manufacturer's protocol (Qiagen RNeasy mini kit, Qiagen; Hilden, Germany). RNA was then reverse transcribed to cDNA following the manufacturer's protocol (QuantiTect Reverse Transcription Kit; Qiagen) and mixed with SYBR Green (PowerUp™ SYBR™ Green Master Mix; Thermo Fisher). For all experiments, gene expression levels were normalized to the housekeeping gene RPL32 expression level and compared to the levels of MC at day 0. All qRT-PCR reactions were performed with $n = 3$ experimental repeats in triplicate, using a Bio-Rad CFX96 thermal-cycler. Heat Map (Supplementary Fig. 2) generated via Morpheus Software (https://software.broadinstitute.org/morpheus). Color varies from Red to Green with red indicating expression the lowest expression and green the highest expression. Each row is independently assessed.

**Tube formation assay.** Differentiated MCs on day 14 were trypsinized and re-plated on growth factor reduced Matrigel in EGM2 with 20% PRP and 50 ng/ml VEGF and cultured for 24–48 h. Wells were fixed in 4% paraformaldehyde, permeabilized with 1% Triton-X100 in PBS, and blocked with 5% goat serum in PBS prior to immunocytochemistry.

**LDL uptake.** Differentiated MCs on day 14 were incubated with acetylated LDL according to the manufacturer's protocol. Cells were exposed to DiI labeled LDL for 4 h, washed with PBS three times, then fixed in 4% paraformaldehyde before imaging and quantification.

**Quantification of NO production.** NO production was measured as a function of NO₂ concentration. Media from differentiated cells under static and shear were collected and assessed for nitrite using the Griess colorimetric reagent (Thermo Fisher).

**Vascular grafts and animal implantations.** Preparation of the SIS vascular grafts with immobilized heparin and SIS and implantations into the carotid arteries of sheep was done as previously described[30]. All animal surgical procedures and other protocols were approved by the Institutional Animal Care and Use Committee at the State University of New York at Buffalo.

**Immunohistochemistry and immunocytochemistry.** Explanted A-TEVs and native carotid arteries were cleaned using saline and pressure fixed in 10% Formalin. Samples were then dehydrated in a series of graded ethanol solutions, xylene substitutes and then embedded in paraffin. Tissue sections (10 μm each) were deparaffinized and subjected to pressure-activated high temperature antigen retrieval. Paraffin sections were first blocked with 5% (v/v) goat serum in PBS. On day 14 and day 20 (after shear) cells were fixed in 4% paraformaldehyde, permeabilized in 0.1% Triton X-100, and blocked in 5% goat serum. Tissue sections and cells were then further incubated with the following primary antibodies: anti-VEGFR1 (1:100, Thermo Fisher), anti-VEGFR2 (1:100, Thermo Fisher Scientific), anti-smooth muscle alpha actin (1:200, Thermo Fisher), anti-CD144 (1:50, Cell Signaling Technologies, Danver, MA), anti-CD16 (1:200, Abgent, San Diego, CA), anti-CD14 (1:100, Abgent), anti-CD38 (1:200, Abcam, Cambridge, MA), anti-EGR2 (1:100, Abcam), anti-CD31 (1:200, Thermo Fisher), anti-phosphorylated-eNOS (1:500, BD Biosciences) in 5% (v/v) goat serum in PBS overnight at 4 °C. Following three washes tissue sections and fixed cells were incubated with Alexa-Fluor conjugated secondary antibodies (1:200 in 5% (v/v) goat serum, Thermo-Fisher) for 1 h at RT. Nuclei were counterstained with Hoechst 33342 (1:200 in PBS, Thermo Fisher) for 5 min at room temperature and images were obtained with a Zeiss Axio Imager microscope (Carl Zeiss GmbH, Jena, Germany).

**Single-cell RNA library preparation and sequencing.** Suspended cells (MC, MCEC, HCAEC) were delivered to the UB Genomics and Bioinformatics Core (UBGBC) for cell counting on the Logos Biosystems LUNA II cell counter using

0.4% Trypan Blue for cell viability. If needed, the cells were diluted to 700-1,000 cells/µl in condition media or 1X PBS containing 0.04% BSA. Once diluted, ~5000 cells were captured on the 10X Genomics Chromium platform using the 3′ transcriptome protocol (V3). After confirmation of efficient cDNA synthesis, samples were processed for Illumina sequencing and quality checked using the Agilent Fragment Analyzer and Qubit fluorescence (Invitrogen). Libraries were pooled to 10 nM and final concentrations were determined using the Kapa Biosystems Universal qPCR system. Pooled libraries were diluted and denatured to 250 pM and run on the NovaSeq 6000 SP flow cell (28 × 91).

**Bioinformatic analysis**. Output from 10X Genomics Cellranger v3.0.1 pipeline was used as input into the R analysis package Seurat[76]. Cells with high unique feature counts, high mitochondrial transcript counts, and high ribosomal transcript counts were filtered from the analysis. The data was normalized using Seurat's LogNormalize, with a scale factor of 10,000. Integration of all three datasets was performed using Seurat's FindIntegrationAnchors followed by IntegrateData. Principal Component Analysis (PCA), cluster analysis and Uniform Manifold Approximation and Projection (UMAP) dimensionality reduction were applied to the integrated dataset.

**Statistics and reproducibility**. All experiments were independently repeated at least three times using different biological samples, and each assay was done with triplicate samples. All results were reproducible during independent biological repeats. Immunofluorescence images were analyzed using ImageJ (NIH) and more than 300 cells were counted in five randomly selected fields of view per image. All plots represent the mean ± the standard deviation. Statistical significance was calculated in Graphpad-PRISM using the following tests: one-way ANOVA followed by Tukey post-hoc test or Dunnett's multiple comparisons test, two-way ANOVA using Sidak's multiple comparisons test. For comparing two conditions only, we used the paired Student's *t*-test.

**Reporting summary**. Further information on research design is available in the Nature Research Reporting Summary linked to this article.

## Data availability
All relevant raw data (Figs. 2b, c, e, f, g, 4c, e, 7a, 8a, and 9) is provided as a source data excel sheet. The RNAseq data discussed in this publication have been deposited in NCBI's Gene Expression Omnibus[77] and are accessible through GEO Series accession number GSE143353.

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

## Acknowledgements

This work was supported by grants from the National Institutes of Health (R01 HL086582) to S.T.A. and F31 HL134323 fellowship to R.J.S.

## Author contributions

R.J.S. and S.T.A. conceived the idea, designed the experiments and analyzed the data. R.J.S. and B.N. performed all experiments. R.J.S. performed all data analysis and prepared the figures. J.K., D.Y., and J.B. performed scRNA-seq and subsequent bioinformatic analysis. D.S. performed in vivo implantations, pre- and post-op animal care and monitoring. R.J.S. and S.T.A. wrote the manuscript.

## Competing interests

S.T.A. and D.D.S. declare financial interest in Angiograft, LLC. The remaining authors declare no competing interests.
