## [Peer Review File · Nature Communications]

Reviewers' comments:

Reviewer #1 (Remarks to the Author):

This manuscript seeks to determine the source of endothelial cells which populate an implanted acellular vascular graft. Overall, the manuscript is well written with an extensive characterization of this potential cell population. The impact of this work is high. However it is more likely to have an impact in the area of ECFC research, rather than tissue engineered vascular grafts due to the specificity of the capture method. The motivation of the paper is the determination of the source of cells, which endothelialize the implanted acellular vascular grafts. Thus the structure of the paper should be reorganized with the in vivo data first to motivate the full in vitro characterization. Additionally this is important as the definitive proof for the type of cells populating the in vivo vascular grafts is not provided.

Specific comments:

Major

1. Overall, the motivation of the paper is the determination of the source of cells, which endothelialize the implanted acellular vascular grafts. Thus the structure of the paper should be reorganized with the in vivo data first. Additionally this is important as the definitive proof for the type of cells populating the in vivo vascular grafts is not provided. Thus, page, 16, lines 354-355, is not fully supported.
2. The importance and impact of the paper is in the field of ECFCs. The paper should emphasize this in terms of this paper: Smadja DM, Melero-Martin JM, Eikenboom J, Bowman M, Sabatier F, Randi AM. Standardization of methods to quantify and culture endothelial colony-forming cells derived from peripheral blood: Position paper from the International Society on Thrombosis and Haemostasis SSC. *J Thromb Haemost.* 2019 Jul;17(7):1190-1194. doi: 10.1111/jth.14462. Epub 2019 May 22. The discussion should include putting this work in context to the ECFC literature.
3. When studying cell capture, the important parameter is shear rate and when studying the effect of flow forces on attached cells, then shear stress is the important parameter. These should be reported accordingly.
4. In the results, on page 7, lines 146-148, there was a low percentage of VEGFR2+ cells (2.56%) on FN. Additionally, the results for CD49 should be reported.
5. The progression of gene expression is highly impactful data. A minor suggestion is to group the artery vs vein genes together (HEY1/2, EphB2/4) for ease of reporting.
6. While the flow cytometry had isotype controls, the IF images did not. This should be included for IF and for the acLDL and matrigel assays.
7. For the shear conditioned MC-EC studies, static controls for the same time period should be included.
8. Importantly, the statistics are incorrect. A student's t-test cannot be used with multiple conditions. A biostatistician should be consulted.
9. Figures: The figures 6 and 8, when individual and merged channel are presented, the immunostained markers should be included in each of the images of the top row, since pink and red are difficult to distinguish. In figure 7, the goal is to present the effect of shear on the cells, thus the x-axis for B-K should be the same as in A. Additionally, in figure 7, the images should be rotated to make the direction of flow uniform. In figure 8, the CD144 appears to label in the

internal elastic lamina due to its autofluorescence. The presentation of controls would likely support this.

Minor

1. The term acellular should be consistently spelled acellular and not a-cellular.
2. In the abstract, page 2, line 39, the term 'right conditions' is vague and should not be used.
3. In the Introduction, the difference between decellularized and devitalized is unclear. It should be distinguished or just one term used.
4. On page 4, line 79, because the anastomotic endothelialization is a potential cells source, the length of the vascular grafts should be included here.
5. In the results, on page 11, line 238, it is reported that phosphorylate eNOS is shown, but the methods reports an antibody to eNOS. This should be corrected.
6. On page, 13, line 289, endothelialization is misspelled.
7. On page 15, line 335, the 'expected response' to what should be clarified. O₂ should be O₂. Also in one place the cell culture condition is listed as 10% CO₂ (page 20, line 448) and in another it is 5% (line 548). Please correct.
8. On page 15, line 341, check reference formatting.
9. On page 17, line 388, in vivo should be italicized.
10. On page 18, line 396, should include references, and line 397, M2-e should be M2e.
11. In author contributions, the Swartz contribution should be included.
12. The methods should include the conditions of PBMCs on the surfaces an when the FBS is replaced by PRP (pg 21, lines 464-5)
13. In Methods, the two sections on page 22, lines 487-495 and page 23, line 513-520, should be better distinguish as both report cell proliferation. Importantly, the concentrations (line 491) do not match the figure 1C results.
14. In references, there are errors on lines 1036, 1043, and 1059.

Reviewer #2 (Remarks to the Author):

Current manuscript by Smith et al implanted SIS immobilized with heparin and VEGF to arterial system and studied the phenotype of attached cells for three months. Additionally, they developed a microfluidic system to simulate native environment in vitro and studied the behavior of whole blood and/or macrophage in this system. They concluded that macrophage may directly contribute to the endothelialization in the tissue-engineered blood vessel both in vitro and in vivo. This is an interesting and well-written manuscript. However, more proof of macrophage differentiation to endothelium or intermediate phenotype is absolutely needed to enhance the rigor of the data.

Major comments:

1. The authors used most of the figures to show macrophage may undergo differentiation towards endothelial phenotype. Many people showed previously that monocyte/macrophage isolated from human peripheral blood or apheresis product can be cultured to endothelial progenitor cells or endothelial colony forming cells in later stage (~ day 14 – 21)(many reports from Dr. Mervin Yoder's group). However, currently there is no paper directly mention that these endothelial phenotype are definitively coming from macrophage or from resident endothelium. Since mixed population of macrophage/endothelial cells are cultured under endothelial culture medium, only endothelial cells will grow and eventually will dilute the macrophage/monocyte population. Additionally, because qRT-PCR only measured the bulk RNA sample, it is no surprise that endothelial phenotype is increasing in qPCR readings. How do you know your endothelial cells at later stage in the in vitro system are actually from macrophage but not contaminated endothelial cells at the beginning? More data such as lineage tracing, or single cell rna seq may be helpful.
2. The microenvironmental cues, in terms of shear stress, oxygen tension, substrate stiffness, cytokines, cellular components, ECM components, and etc. are not comparable between in vitro

platforms as compared to in vivo conditions. Why do you think your current platform can be directive for in vivo conditions?

3. In figure 8, the authors showed that after 1 month, endothelium appeared in the tissue-engineered vessel possibly due to macrophage differentiation, which is similar to what they saw in in vitro platform. How do you know endothelium is resulted from differentiation of macrophage? Can you rule out the possibility that macrophage-secreted factors attract endothelium to adhere and grow? Again if the authors would like to conclude this, lineage tracing should be done.

4. What is the rationale of using HUVEC as a vein EC control? It is well known that endothelial cells display organotypicity and HUVECs are isolated from veins in umbilical cord, which may not be a good control for adult EC markers. Also it would be helpful if the authors provide artery endothelium control.

Minor comments:

1. In figure 3, it is really hard to tell the difference of cell size by immunostaining images, please use zoom in images.

2. Spelling typo: line 642 "passed though" should be "passed through".

3. Line 586 – 588: NO production: Griess assay is to measure nitrite (NO₂) concentration not nitrate (NO₃). Please correct.

Reviewer #3 (Remarks to the Author):

This manuscript describes the potential role of monocytes in the endothelialization of vascular grafts, and in particular focuses on the potential mechanism of endothelialization of decellularized tissue grafts with VEGF165 immobilized on the graft surface in a large animal ovine model. The authors demonstrate binding of monocytes to VEGF and present a novel protocol to promote monocyte proliferation and differentiation into an endothelial cell (EC)-like phenotype in vitro. The authors present compelling evidence to suggest that monocytes can indeed bind VEGF immobilized on the surface and that under specific conditions these cells can differentiate toward an EC-like phenotype. This is not surprising, as other studies have demonstrated monocyte differentiation to endothelial cells and the role of CD14⁺ cells in re-endothelialisation following balloon injury has been demonstrated. In fact, these studies should be described in both the Introduction and further discussed/acknowledged in the Discussion.

This evidence will be of use to our growing understanding of various phenotypes monocytes can differentiate toward and our understanding of small diameter graft endothelialization. However, while this work lends strong support for monocyte dependent mechanism of endothelialization, it does not directly prove it and this would admittedly be difficult to do in large animal models where it is difficult to obtain directly labelled monocytes or irradiate bone marrow. A discussion of whether similar mechanisms have been observed or are likely to occur in small animal (rodent) models may be useful to inform further mechanistic studies.

However, it would be useful to the field to demonstrate in large animal models that these effects are VEGF dependent as claimed in this article. It is not clear from the experimental methods and the results section what controls were employed in the animal model and if these effects are only observed when monocytes are captured by VEGF via their VEGFR1 receptor as proposed here. It is also not clear if these are the conditions necessary for monocytes to differentiate to EC-like cells or is something unique about decellularized tissue, the ovine model or similar? In fact, If fibronectin captures the same population of cells as VEGF (CD14⁺, VEGFR1 expressing), would similar mechanism of endothelialization be expected on fibronectin-coated grafts and how does this correlate with the literature on fibronectin-coated surfaces? The authors discussed potential limitations of fibronectin non-specificity and lack of endothelialization in the graft center in the Introduction, but only in relation to scarcely present EPCs, not to monocytes.

Minor corrections/clarifications are outlined below:

Please define SIS abbreviation in the abstract

Line 54 (Introduction)- the statements about the utility and testing of decellularized tissues

showing potential in pre-clinical and clinical trials needs to be a little more detailed; how many of these constructs have been in clinical trials and what were the outcomes. The long reference list (1-13) should be split by at least pre-clinical vs clinical trial studies.

Line 121- it is stated that all captured cells expressed CD31, but only ~70% of cells at 15 dyn/cm² did

Response to Reviewers

1. Reviewer 1:

1. Overall, the motivation of the paper is the determination of the source of cells, which endothelialize the implanted acellular vascular grafts. Thus the structure of the paper should be reorganized with the in vivo data first. Additionally this is important as the definitive proof for the type of cells populating the in vivo vascular grafts is not provided. Thus, page, 16, lines 354-355, is not fully supported.

Ans: In this study we present the *in-vivo* data last as a means of indicating the significance of the finding. We respectfully disagree with the organization of the paper, but ultimately leave this to the editor to determine which sequence is best.

Additionally, we have included two major experiments to attempt to clarify the source of the cells. While we cannot prove the source of EC in-vivo, we can demonstrate that in-vitro there is no contaminating EC during the differentiation process. We achieved this by two means. The first we performed was single cell RNA sequencing of three populations of cells. MC isolated from fresh blood, HCAEC cultured under high shear and differentiated MCEC/M2e cells cultured under high shear. The results indicate that the initial population of cells (MC) were distinct from HCAEC with NO cells expressing the EPC markers VEGFR2 and AC133. In contrast, the resulting MCEC/M2e population was significantly similar to mature HCAEC while expressing some macrophage genes. Please see the new Figure 8.

The second experiment was the development of a fluorescent reporter tied to puromycin selection. MC were transduced with a lentivirus contained a previously reported CD68 promoter sequence fused to ZsGreen combined with an IRES-Puromycin. This enabled ZsGreen tracking of cells as well as selection of cells expressing ONLY CD68 via puromycin selection. EC do not express CD68, thus any contaminating EC would be removed via selection. Differentiation was then carried out as described and the CD68-ZsGreen marker remained expressed throughout and co-stained with EC markers CD144 and VEGFR2 and MC marker CD14. Furthermore, CD68 expression decreased after differentiated cells were subjected to shear. Please see new Figure 9 for immunocytochemistry. As a control, HCAEC cells were also transduced with the same vector but all cells died upon puromycin selection, suggesting that CD68-Pr was not active in EC. To further emphasize this point, HCAEC were transduced with a dual promoter lentiviral vector (pCD68-LVDP) encoding for ZsGreen under the CD68-Pr and DsRed2 under the PGK promoter. This vector was developed in our laboratory and contains insulator and terminator sequences that diminish promoter interference. After transduction and puromycin selection of MC with pCD68-ZsG-Puro, all cells were ZsGreen+ indicating active CD68-Pr. Transduced HCAEC expressed DsRed but not ZsGreen, in agreement with lack of CD68-Pr activity seen with puromycin selection.

Finding a definitive source of the cells in-vivo would require either transgenic sheep or a completely different (and therefore less physiologically relevant) animal model.

2. The importance and impact of the paper is in the field of ECFCs. The paper should emphasize this in terms of this paper: Smadja DM, Melero-Martin JM, Eikenboom J, Bowman M, Sabatier F, Randi AM. Standardization of methods to quantify and culture endothelial colony-forming cells derived from peripheral blood: Position paper from the International Society on Thrombosis and Haemostasis SSC. J Thromb Haemost. 2019 Jul;17(7):1190-1194. doi: 10.1111/jth.14462. Epub 2019 May 22. The discussion should include putting this work in context to the ECFC literature.

Ans: We have reviewed the suggested paper and have concluded that our protocol is significantly different than the protocol suggested within the Smadja et al paper. However, we have included a brief statement regarding this difference within the discussion and is noted in red. Furthermore, and as discussed above- we have included a lentiviral transduction experiment wherein cells are selected by puromycin on the basis of monocyte/macrophage specific expression of CD68- thereby removing any potential ECFC/EPCs from the culture.

3. When studying cell capture, the important parameter is shear rate and when studying the effect of flow forces on attached cells, then shear stress is the important parameter. These should be reported accordingly.

We report shear stress as dyn/cm^2 and present the accompanying flow rates in Figure 1A for cell capture. For studying shear stress on attached cells we indicate shear stress (1-10 dyn/cm^2) and in the methods section we indicate the RPM required to induce these forces.

4. In the results, on page 7, lines 146-148, there was a low percentage of VEGFR2+ cells (2.56%) on FN. Additionally, the results for CD49 should be reported.

Ans: Although we do see a very small percentage of VEGFR2+ cells after adherence on FN surfaces, it should be noted that we did not see any colony forming units or contaminating EC cells at both pre and post-shear time points. Additionally, we have included additional experiments wherein cells transduced with a virus containing a vector with the CD68 promoter driving ZsGreen expression along with an IRES-Puro selection marker, we remove any possible contaminating cells by selecting cells with puromycin from day 3 onwards. Any EC cell contamination is removed by puromycin selection, as EC do not express CD68. * Please see response to comment 1 for full details on this new result and Figure 9.

Additionally, we have removed CD49 from the figure, as it does not contribute to this study.

5. The progression of gene expression is highly impactful data. A minor suggestion is to group the artery vs vein genes together (HEY1/2, EphB2/4) for ease of reporting.

Ans: We appreciate the suggestion, and have rearranged the graphs of Hey1/2 EphB2/B4 in Figure 7.

6. While the flow cytometry had isotype controls, the IF images did not. This should be included for IF and for the acLDL and matrigel assays.

Ans: We have included isotype control/secondary alone controls in supplemental materials (Figure SI-3). For acLDL there was no staining using antibodies- the acLDL was directly labelled as per manufacturer's kit.

7. For the shear conditioned MC-EC studies, static controls for the same time period should be included.

Ans: We have included static controls and have clarified the time points. Sheared cells were sheared on Day 9 to end on Day 14 to be consistent with static time points.

8. Importantly, the statistics are incorrect. A student's t-test cannot be used with multiple conditions. A biostatistician should be consulted.

Ans: We have updated this section to report student t-test and ANOVA for multiple condition statistics. T-test was only used for comparisons between 2 conditions and ANOVA for anything beyond 2 conditions.

9. Figures: The figures 6 and 8, when individual and merged channel are presented, the immunostained markers should be included in each of the images of the top row, since pink and red are difficult to distinguish. In figure 7, the goal is to present the effect of shear on the cells, thus the x-axis for B-K should be the same as in A. Additionally, in figure 7, the images should be rotated to make the direction of flow uniform. In figure 8, the CD144 appears to label in the internal elastic lamina due to its autofluorescence. The presentation of controls would likely support this.

Ans: Figures 6 and 9 have been modified to include individual channel labels to increase clarity. Figure 7 has been updated with new plots for A-K with the same x-axis as well as new arterial EC controls. In Figure 7, the images after shear are difficult to reposition for imaging. Rotation of the image to make the direction of flow uniform would cause cropping and size issues. The inclusion of arrows to indicate direction of flow was included to help clarify the direction of flow. In Figure 10, the white arrow points to the elastic lamina and is now indicated in Figure captions.

Minor

1. The term acellular should be consistently spelled acellular and not a-cellular.

Ans: Changes made and indicated in red.

2. In the abstract, page 2, line 39, the term 'right conditions' is vague and should not be used.

Ans: Changes made and indicated in red. Changed to “Using an optimized protocol” to be less vague.

3. In the Introduction, the difference between decellularized and devitalized is unclear. It should be distinguished or just one term used.

Ans: Changes made and indicated in red. As there is very little difference between devitalized and decellularized, we used only decellularized to be more clear.

4. On page 4, line 79, because the anastomotic endothelialization is a potential cells source, the length of the vascular grafts should be included here.

Ans: Changes made and indicated in red. The length of the grafts is now indicated (5cm long).

5. In the results, on page 11, line 238, it is reported that phosphorylate eNOS is shown, but the methods reports an antibody to eNOS. This should be corrected.

Ans: Changes made and indicated in red. We have updated the methods to correctly report the use of anti-phosphorylated eNOS.

6. On page, 13, line 289, endothelialization is misspelled.

Ans: Changes made and indicated in red.

7. On page 15, line 335, the ‘expected response’ to what should be clarified. O₂ should be O₂. Also in one place the cell culture condition is listed as 10% CO₂ (page 20, line 448) and in another it is 5% (line 548). Please correct.

Ans: Changes made and indicated in red. We use 5% CO₂. The “expected response” was changed to “literature supported, wound mediated” response. Further down in the paragraph we provide references.

8. On page 15, line 341, check reference formatting.

Ans: Changes made and indicated in red.

9. On page 17, line 388, in vivo should be italicized.

Ans: Changes made and indicated in red.

10. On page 18, line 396, should include references, and line 397, M2-e should be M2e.

Ans: Changes made and indicated in red. References were added.

11. In author contributions, the Swartz contribution should be included.

Ans: Changes made and indicated in red.

12. The methods should include the conditions of PBMNs on the surfaces an when the FBS is replaced by PRP (pg 21, lines 464-5)

Ans: Changes made and indicated in red.

13. In Methods, the two sections on page 22, lines 487-495 and page 23, line 513-520, should be better distinguish as both report cell proliferation. Importantly, the concentrations (line 491) do not match the figure 1C results.

Cellular proliferation reported in Figure 1C is described in methods under the heading “Proliferation on VEGF functionalized surface. We have added a reference to Figure 1C under this heading. The “biological activity of recombinant VEGF” section has been removed.

14. In references, there are errors on lines 1036, 1043, and 1059.

Ans: References have been fixed.

2. Reviewer #2 (Remarks to the Author):

Current manuscript by Smith et al implanted SIS immobilized with heparin and VEGF to arterial system and studied the phenotype of attached cells for three months. Additionally, they developed a microfluidic system to simulate native environment in vitro and studied the behavior of whole blood and/or macrophage in this system. They concluded that macrophage may directly contribute to the endothelialization in the tissue-engineered blood vessel both in vitro and in vivo. This is an interesting and well-written manuscript. However, more proof of macrophage differentiation to endothelium or intermediate phenotype is absolutely needed to enhance the rigor of the data.

Major comments:

1. The authors used most of the figures to show macrophage may undergo differentiation towards endothelial phenotype. Many people showed previously that monocyte/macrophage isolated from human peripheral blood or apheresis product can be cultured to endothelial progenitor cells or endothelial colony forming cells in later stage (~ day 14 – 21)(many reports from Dr. Mervin Yoder's group). However, currently there is no paper directly mention that these endothelial phenotype are definitively coming from macrophage or from resident endothelium. Since mixed population of macrophage/endothelial cells are cultured under endothelial culture medium, only endothelial cells will grow and eventually will dilute the macrophage/monocyte population. Additionally, because qRT-PCR only measured the bulk RNA sample, it is no surprise that endothelial phenotype is increasing in qPCR readings. How do you know your endothelial cells at later stage in the in vitro system are actually from macrophage but not contaminated endothelial cells at the beginning? More data such as lineage tracing, or single cell rna seq may be helpful.

Ans: Though it is theoretically possible to isolate circulating EPCs or circulating EC from the blood, it is exceedingly rare to isolate and culture EPCs from healthy human donors. However, it is important to note that we use both FN or VEGF surfaces to isolate cells on the surface for 1hr and then wash the cells to remove unbound cells. Current methods to culture EPCs or EC from the blood require this initial selection step to remove monocytes/macrophages and *replating* unbound cells on ECM coated dishes for additional 3+ days before EC/EPC attachment occurs. Thus with our protocol the likelihood of EC contamination is very low. In addition, we observe the cells daily and do not find any EC colony contamination. Furthermore, the immunocytochemistry performed on Day 14 under both static and shear conditions shows cells with CD14/CD16/CD163 staining- markers that are not present on EC or EPC cells. All cells at day 14 with or without shear have these markers.

The first we performed was single cell RNA sequencing of three populations of cells. MC isolated from fresh blood, HCAEC cultured under high shear and differentiated MCEC/M2e cells cultured under high shear. The results indicate that the initial population of cells (MC) were distinct from HCAEC with NO cells expressing the EPC markers VEGFR2 and AC133. In contrast, the resulting MCEC/M2e population was

significantly similar to mature HCAEC while expressing some macrophage genes. Please see the new Figure 8.

The second experiment was the development of a fluorescent reporter tied to puromycin selection. MC were transduced with a lentivirus contained a previously reported CD68 promoter sequence fused to ZsGreen combined with an IRES-Puromycin. This enabled ZsGreen tracking of cells as well as selection of cells expressing ONLY CD68 via puromycin selection. EC do not express CD68, thus any contaminating EC would be removed via selection. Differentiation was then carried out as described and the CD68-ZsGreen marker remained expressed throughout and co-stained with EC markers CD144 and VEGFR2 and MC marker CD14. Furthermore, CD68 expression decreased after differentiated cells were subjected to shear. Please see new Figure 9 for immunocytochemistry. As a control, HCAEC cells were also transduced with the same vector but all cells died upon puromycin selection, suggesting that CD68-Pr was not active in EC. To further emphasize this point, HCAEC were transduced with a dual promoter lentiviral vector (pCD68-LVDP) encoding for ZsGreen under the CD68-Pr and DsRed2 under the PGK promoter. This vector was developed in our laboratory and contains insulator and terminator sequences that diminish promoter interference. After transduction and puromycin selection of MC with pCD68-ZsG-Puro, all cells were ZsGreen+ indicating active CD68-Pr. Transduced HCAEC expressed DsRed but not ZsGreen, in agreement with lack of CD68-Pr activity seen with puromycin selection.

2. The microenvironmental cues, in terms of shear stress, oxygen tension, substrate stiffness, cytokines, cellular components, ECM components, and etc. are not comparable between in vitro platforms as compared to in vivo conditions. Why do you think your current platform can be directive for in vivo conditions?

Ans: While it is impossible to replicate in vivo conditions in vitro we believe our conditions are more comparable to in vivo conditions when comparing our culture conditions to other literature. The inclusion of autologous serum through PRP at a high percentage (v/v) is an attempt to recapitulate what the cells experience in the blood. Additionally, the inclusion of shear is a major component of our platform- as cells in vivo will experience this force. Other reported methods have not used shear on monocyte derived cells. We do agree that while it is impossible to replicate in vivo conditions, we do however believe our conditions to be closer to in vivo conditions than other reported methods and continue to work on improving conditions to become more physiologically relevant. It is important to note however, that even though we cannot replicate in vivo conditions, we demonstrate similar end results. After 1mo in vivo, the lumen is coated with cells that express both MC and EC markers- similar to that observed after our in vitro protocol.

3. In figure 8, the authors showed that after 1 month, endothelium appeared in the tissue-engineered vessel possibly due to macrophage differentiation, which is similar to what

they saw in in vitro platform. How do you know endothelium is resulted from differentiation of macrophage? Can you rule out the possibility that macrophage-secreted factors attract endothelium to adhere and grow? Again if the authors would like to conclude this, lineage tracing should be done.

Ans: While we cannot rule out that macrophage secreted factors could have attracted EC and EPCs to repopulate the graft, it should be noted that we present immunohistochemistry of the lumen wherein the graft lumen is populated with cells expressing BOTH MC and EC markers. EC do not express CD14 or CD163- yet the lumen of our grafts contain an endothelium with cells expressing both these markers alongside classic EC markers eNOS and CD144. Our study was performed in sheep, a physiologically similar animal model to humans. Lineage tracing in a sheep is currently not a feasible option. Lineage tracing in other models, while possible, is not physiologically similar to humans. It is well established that mice/rat EC rapidly proliferate from anastomotic sites. However, it should be noted that while not presented in this study, our early study published in FASEB using mice, presented similar findings on VEGF grafts after 1month. Luminal cells expressed both MC and EC markers.

4. What is the rationale of using HUVEC as a vein EC control? It is well known that endothelial cells display organotypicity and HUVECs are isolated from veins in umbilical cord, which may not be a good control for adult EC markers. Also it would be helpful if the authors provide artery endothelium control.

Ans: We have included an arterial EC control for gene analysis, single cell RNA seq, ac-LDL, and Nitrite production.

Minor comments:

1. In figure 3, it is really hard to tell the difference of cell size by immunostaining images, please use zoom in images.

Ans: The images are representative, please see the quantification chart for clarity on sizes, we have also included representative magnified images.

2. Spelling typo: line 642 “passed though” should be “passed through”.

Ans: We have corrected this and indicated the correction in red.

3. Line 586 – 588: NO production: Griess assay is to measure nitrite (NO₂) concentration not nitrate (NO₃). Please correct.

Ans: We have corrected this and indicated the correction in red.

3. Reviewer #3 (Remarks to the Author):

This manuscript describes the potential role of monocytes in the endothelialization of

vascular grafts, and in particular focuses on the potential mechanism of endothelialization of decellularized tissue grafts with VEGF165 immobilized on the graft surface in a large animal ovine model. The authors demonstrate binding of monocytes to VEGF and present a novel protocol to promote monocyte proliferation and differentiation into an endothelial cell (EC)-like phenotype in vitro.

The authors present compelling evidence to suggest that monocytes can indeed bind VEGF immobilized on the surface and that under specific conditions these cells can differentiate toward an EC-like phenotype. This is not surprising, as other studies have demonstrated monocyte differentiation to endothelial cells and the role of CD14+ cells in re-endothelialisation following balloon injury has been demonstrated. In fact, these studies should be described in both the Introduction and further discussed/acknowledged in the Discussion.

Ans: We have now included acknowledgment of this study in the discussion and is indicated in red.

This evidence will be of use to our growing understanding of various phenotypes monocytes can differentiate toward and our understanding of small diameter graft endothelialization. However, while this work lends strong support for monocyte dependent mechanism of endothelialization, it does not directly prove it and this would admittedly be difficult to do in large animal models where it is difficult to obtain directly labelled monocytes or irradiate bone marrow. A discussion of whether similar mechanisms have been observed or are likely to occur in small animal (rodent) models may be useful to inform further mechanistic studies.

Ans: We have performed similar work in a mouse model (FASEB 2019). Added to discussion: “However, we have previously demonstrated similar results in a mouse model, whereby VEGF graft lumens were comprised of cells expressing both EC and MC/MΦ markers and that this was dependent upon VEGF. Grafts lacking VEGF, lacked proper remodeling of both endothelium and medial layers”

However, it would be useful to the field to demonstrate in large animal models that these effects are VEGF dependent as claimed in this article. It is not clear from the experimental methods and the results section what controls were employed in the animal model and if these effects are only observed when monocytes are captured by VEGF via their VEGFR1 receptor as proposed here.

Ans: For sheep implantations, our control was grafts without VEGF- in this model, the control grafts occluded within hours of implantation and thus without VEGF grafts failed. However, as noted above- in mouse models, grafts without VEGF did not fail (likely due to the significantly higher flow rate in mouse abdominal aorta compared to the carotid artery of sheep. VEGF grafts exhibited similar endothelium regeneration as seen in sheep. In comparison, heparin only grafts failed to regenerate proper endothelium or medial layers- along with a predominantly inflammatory response- the heparin only grafts were clearly insufficiently remodeled compared to VEGF grafts.

It is also not clear if these are the conditions necessary for monocytes to differentiate to EC-like cells or is something unique about decellularized tissue, the ovine model or similar? In fact, If fibronectin captures the same population of cells as VEGF (CD14+, VEGFR1 expressing), would similar mechanism of endothelialization be expected on fibronectin-coated grafts and how does this correlate with the literature on fibronectin-coated surfaces? The authors discussed potential limitations of fibronectin non-specificity and lack of endothelialization in the graft center in the Introduction, but only in relation to scarcely present EPCs, not to monocytes.

Ans: It was surprising to see that FN and iVEGF surfaces conferred similar differentiation potential to the cells, likely indicating that *immobilized* VEGF is not required. However, FN surfaces (and iVEGF) are cultured in the presence of soluble VEGF. In addition, the use of PRP (rich in Fibronectin AND VEGF) further diminishes the difference between FN and iVEGF surfaces. It is possible though that iVEGF may confer greater EC differentiation capacity under flow conditions- something we are unable to fully test in-vitro as the cells (so far) require a period of static adherence and spreading prior to shear.

In addition, FN has been used in animal models as indicated in the Intro/Discussion. FN grafts occluded in the center resulting in incomplete endothelialization. Likely due to the non-specificity of RGD ligand contained within fibronectin, a ligand that binds a large variety of cells including platelets.

Minor corrections/clarifications are outlined below:
Please define SIS abbreviation in the abstract

Ans: We have corrected this and indicated the correction in red.
Line 54 (Introduction)- the statements about the utility and testing of decellularized tissues showing potential in pre-clinical and clinical trials needs to be a little more detailed; how many of these constructs have been in clinical trials and what were the outcomes. The long reference list (1-13) should be split by at least pre-clinical vs clinical trial studies.

Ans: We have split the references as requested and indicated the correction in red.
Line 121- it is stated that all captured cells expressed CD31, but only ~70% of cells at 15 dyn/cm² did

Ans: We have corrected this and indicated the correction in red.

Reviewers' comments:

Reviewer #1 (Remarks to the Author):

The authors were responsive to most of the critiques in the prior reviews. They performed additional experiments of both single cell RNA sequencing and MC-promoter selection on the in vitro differentiation of the MCECs.

Unfortunately, as the authors state in the reviewer response, "Finding a definitive source of the cells in vivo would require either transgenic sheep or a completely different (and therefore less physiologically relevant) animal model." Thus the claim that the endothelialization of the acellular grafts in vivo results from MCs is not proven. The authors cite the difficulty of transgenic sheep, but also have access to a mouse model in which transgenic lineage tracing can be done. If this were added to the manuscript then the reported claims would be supported.

Reviewer #3 (Remarks to the Author):

While the authors have addressed reviewer's comments in the rebuttal, the comments are not always addressed in the manuscript. The comments (esp discussion points and acknowledgement of previous studies/limitations) is of value to other readers, not just to satisfy the reviewer. It would be useful for the discussion points in the response to reviewers to be described in the manuscript Introduction, Discussion and Methods (in vivo controls).

Response to Reviewers

Reviewer 1:

The authors were responsive to most of the critiques in the prior reviews. They performed additional experiments of both single cell RNA sequencing and MC-promoter selection on the in vitro differentiation of the MCECs.

Unfortunately, as the authors state in the reviewer response, "Finding a definitive source of the cells in vivo would require either transgenic sheep or a completely different (and therefore less physiologically relevant) animal model." Thus, the claim that the endothelialization of the acellular grafts in vivo results from MCs is not proven. The authors cite the difficulty of transgenic sheep, but also have access to a mouse model in which transgenic lineage tracing can be done. If this were added to the manuscript then the reported claims would be supported.

Response: We appreciate the suggestion to do lineage tracing in a mouse model- as this would provide additional evidence of monocyte to endothelial transition and their role in endothelialization. We are looking to do such a study in the near future.

In response to the reviewer and to the editor's recommendation, we have subsequently revised our manuscript such that the ultimate claim of endothelialization from MC in-vivo is presented first and formed the basis for our hypothesis (Figure 1). This is followed by careful and thorough in vitro analysis of MC to EC transition in-vitro that supports our hypothesis. All changes reflecting this re-ordering of the figures are in red text.

Reviewer 3:

While the authors have addressed reviewer's comments in the rebuttal, the comments are not always addressed in the manuscript. The comments (esp discussion points and acknowledgement of previous studies/limitations) is of value to other readers, not just to satisfy the reviewer. It would be useful for the discussion points in the response to reviewers to be described in the manuscript Introduction, Discussion and Methods (in vivo controls).

Response: We have carefully reviewed our previous rebuttal statements and have provided additional points within the introduction and discussion.

Introduction

Second paragraph line 81- "When implanted into the abdominal aorta of a mouse model, the 1mm diameter VEGF grafts were fully endothelialized within one month, consisted of pro-regenerative-anti-inflammatory cells and exhibited distinct vascular remodeling towards the native state³⁶. Furthermore, when implanted into the carotid arteries of a clinically relevant ovine animal model, such small diameter (4.5mm), 5cm long grafts exhibited high patency rates, fully endothelialized within one month, and developed a functional and contractile medial layer by three months post-implantation³⁷⁻³⁹."

→ Here we describe in more detail our previous work using the VEGF grafts in mice and sheep. We provide additional discussion of the controls in the discussion section.

Discussion

Second Paragraph (new) line 372

"VEGF was essential in maintaining patency and promoting remodeling. In the absence of VEGF, grafts failed due to occlusion within hours of implantation in the ovine model, suggesting that heparin alone was not enough to prevent clotting. However, in mouse models, grafts without VEGF did not occlude, likely due to the significantly higher flow rate in mouse abdominal aorta compared to the carotid artery of sheep. However, heparin only grafts exhibited a high degree of inflammation and lacked well-defined endothelial and medial layers. In contrast,

VEGF grafts were fully endothelialized within one month, consisted of pro-regenerative, anti-inflammatory cells and exhibited distinct vascular remodeling towards the native state³⁶.”

- ➔ Here we discuss our previous works, including discussion of the controls (heparin only) as discussed in the prior response to reviewers.

Seventh Paragraph (new) Line 454

“Our results indicate that MC differentiate into EC on both iVEGF and FN surfaces, likely suggesting that immobilized VEGF may not be required. However, MC are coaxed to differentiate into EC in the presence of high concentrations of soluble VEGF and PRP (rich in FN *and* VEGF) on both FN and iVEGF surface, which may be diminishing the need for differentiation signals by the immobilized ligands. In addition, it is possible that iVEGF may confer greater EC differentiation capacity under flow conditions early on during differentiation *in vivo* - something we are unable to fully test *in vitro* as the cells require a period of static adherence and spreading prior to the onset of differentiation. On the other hand, as we show here VEGF grafts do not occlude *in vivo* possibly because they can selectively attract VEGFR1 expressing MC; while FN grafts were shown to occlude^{31,33}, likely because they attract other cell types expressing FN binding integrins, including platelets.”

- ➔ Here we provide additional discussion regarding the use of VEGF to differentiate MC to EC as well as discussion on the role FN plays considering the point raised by the reviewer that VEGF and FN confer similar capture abilities as well as differentiation potential.

REVIEWERS' COMMENTS:

Reviewer #1 (Remarks to the Author):

The authors have responded to all of the reviewer comments.

Minor notes:

Lines 454-464 are in a different font.

Image SI-4 has notes in the title